https://doi.org/10.1038/s41467-021-24366-4　　**OPEN**

# Display of the human mucinome with defined O-glycans by gene engineered cells

Rebecca Nason[1,11], Christian Büll[1,11], Andriana Konstantinidi[1], Lingbo Sun[1], Zilu Ye[1], Adnan Halim[1], Wenjuan Du[2], Daniel M. Sørensen [1], Fabien Durbesson[3], Sanae Furukawa[1], Ulla Mandel [1], Hiren J. Joshi [1], Leo Alexander Dworkin [1], Lars Hansen[1], Leonor David [4,5], Tina M. Iverson [6], Barbara A. Bensing[7], Paul M. Sullam[7], Ajit Varki[8], Erik de Vries[2], Cornelis A. M. de Haan [2], Renaud Vincentelli[3], Bernard Henrissat [1,3,9], Sergey Y. Vakhrushev [1], Henrik Clausen [1✉] & Yoshiki Narimatsu [1,10✉]

Mucins are a large family of heavily O-glycosylated proteins that cover all mucosal surfaces and constitute the major macromolecules in most body fluids. Mucins are primarily defined by their variable tandem repeat (TR) domains that are densely decorated with different O-glycan structures in distinct patterns, and these arguably convey much of the informational content of mucins. Here, we develop a cell-based platform for the display and production of human TR O-glycodomains (~200 amino acids) with tunable structures and patterns of O-glycans using membrane-bound and secreted reporters expressed in glycoengineered HEK293 cells. Availability of defined mucin TR O-glycodomains advances experimental studies into the versatile role of mucins at the interface with pathogenic microorganisms and the microbiome, and sparks new strategies for molecular dissection of specific roles of adhesins, glycoside hydrolases, glycopeptidases, viruses and other interactions with mucin TRs as highlighted by examples.

[1] Copenhagen Center for Glycomics, Departments of Cellular and Molecular Medicine and Odontology, Faculty of Health Sciences, University of Copenhagen, Copenhagen, Denmark. [2] Section Virology, Division of Infectious Diseases and Immunology, Department Biomolecular Health Sciences, Faculty of Veterinary Medicine, Utrecht University, CL Utrecht, the Netherlands. [3] Architecture et Fonction des Macromolécules Biologiques, CNRS, Aix-Marseille Université, Marseille, France. [4] Institute of Molecular Pathology and Immunology of the University of Porto/I3S, Porto, Portugal. [5] Medical Faculty of the University of Porto, Porto, Portugal. [6] Departments of Pharmacology and Biochemistry, Vanderbilt University, Nashville, TN, USA. [7] Department of Medicine, The San Francisco Veterans Affairs Medical Center, and the University of California, San Francisco, CA, USA. [8] The Glycobiology Research and Training Center, and the Department of Cellular and Molecular Medicine, University of California, San Diego, CA, USA. [9] Department of Biological Sciences, King Abdulaziz University, Jeddah, Saudi Arabia. [10] GlycoDisplay ApS, Copenhagen, Denmark. [11]These authors contributed equally: Rebecca Nason, Christian Büll. ✉email: hclau@sund.ku.dk; yoshiki@sund.ku.dk

Nature's overarching solution for fulfilling the need for symbiosis with a vast community of microorganisms— our microbiomes—are mucins[1–3]. Mucins in the gut constitute the primary barrier as well as the ecological niche for the microbiome. Dynamic replenishment of mucin layers provides a constant selection of the resident microbiome through adhesive interactions, and degradation of mucin O-glycans by members of the microbiota supply nutrients[4–6]. Mucins are a large family of heavily glycosylated proteins that line all mucosal surfaces and represent the major macromolecules in body fluids[2]. Mucins clear, contain, feed, direct, and continuously replenish our microbiomes, limiting unwanted co-habitation and repressing harmful pathogenic microorganisms[7]. Recent studies show that specific mucins can disperse biofilms and disrupt bacterial aggregation[8], but the molecular basis of these effects remains largely obscure. Mucin O-glycans present the essential binding opportunities and informational cues for microorganisms via adhesins, however, our understanding of these features is essentially limited to results from studies with simple oligosaccharides without the protein context of mucins and the higher-order features presented by dense O-glycan motifs. Mucins are notoriously difficult to isolate due to their size and heterogeneity, and production by recombinant expression in cell lines is impeded due to difficulties with the assembly of full coding expression plasmids often resulting in heterogeneous products[9].

State-of-the-art technologies to capture the informational content of mucins are confined to studies with synthetic and isolated O-glycans[10], synthetic and chemoenzymatically produced short glycopeptides[11], and synthetic glycopeptides as well as non-natural polymers[12,13]; all of which are rare commodities that do not reflect the complex information captured in distinct human mucins by their display of patterns and structures of O-glycans. With the advent of the facile nuclease-based gene engineering technologies, it has become possible to engineer mammalian cells with combinatorial knockout (KO) and knockin (KI) of glycosylation genes to display subsets and distinct features of the glycome on the cell surface or on secreted reporter proteins in order to probe biological interactions dependent on glycans[14–16]. Genetic engineering provides opportunities for interpretation and dissection of the glycosylation genes, biosynthetic pathways, and structural features required for identifiable interactions with the cell library[17]. Importantly, such cell-based display strategies allow for the presentation of glycans in the natural context of glycoproteins and the cell surface, and this has provided the first experimental evidence for the existence of higher-order binding motifs consisting of O-glycans in dense patterns[15].

There are at least 18 distinct mucin genes encoding membrane or secreted mucins[18]. The large gel-forming secreted mucins may form oligomeric networks or extended bundles through inter- and intramolecular disulfide bridges in the C- and N-terminal cysteine-rich regions[2]. A common characteristic of all mucins is that the major part of their extracellular region comprises a variable number of imperfect tandem repeated (TR) sequences (also called PTS sequences) that carry dense O-glycans (Fig. 1), with the notable exception of MUC16 that contains a large, densely O-glycosylated N-terminal region without TRs[19,20]. Arguably, the main cues for the microbiota lie in the TR regions that display O-glycans of diverse structures and positions in unique patterns. Interestingly, the TR regions appear poorly conserved throughout evolution in contrast to the flanking regions of the large mucins[21], which has been interpreted to reflect that the TR regions simply need to carry dense O-glycans without specific patterns. An alternative interpretation is that the divergence in TR sequences has co-evolved with the microbiota to govern refined interactions with larger motifs of O-glycan patterns as recently suggested for streptococcal serine-rich adhesins[15]. The TR regions of mucins are quite distinct in length and in sequences with a distinct spacing of O-glycosites[22], and TRs in any mucin exhibit individual variability in numbers as well as to some degree in actual sequences[21]. Thus, there are rich opportunities for unique codes in mucin TRs, governed by the particular display of patterns and structures of O-glycans. The mucin TRs and their glycocodes may be considered the informational content of mucins and thus comprise the mucinome. The TR mucinome provides a much greater potential binding epitome than the comparatively limited repertoire of binding epitopes comprised of simple oligosaccharide motifs available in humans[23].

Here, we sought to capture the molecular cues contained in human mucin TRs and enable molecular dissection of these cues. We developed a cell-based platform for the display and production of representative mucin TRs with defined O-glycans. We reasoned that most of the features of human mucin TRs could be displayed in shorter segments of 150–200 amino acids, and used a GFP-tagged expression design containing representative TRs from different mucins to produce a library of the cell membrane and secreted mucin TR reporters in human embryonic kidney (HEK293) cells with distinct programmed O-glycosylation capacities. Strikingly, we found that these mucin TR reporters could readily be produced as highly homogeneous molecules with essentially complete O-glycan occupancies and with distinct O-glycan structures in amounts that enabled us to characterize the simplest reporters by intact mass spectrometry (MS), and hence circumvent the longstanding obstacles with protease digestion and bottom-up analysis of mucins[24,25]. We demonstrate the utility of the cell-based mucin display through probing and dissecting selective mucin TR binding specificities of microbial adhesins, and the power of making defined mucin TRs available through analysis of the substrate specificities of microbial glycopeptidases, as well as by demonstrating selective binding by an influenza virus. These findings widely open the mucin and microbiome fields for studies with libraries of homogeneous mucin TR glycodomains, and for entirely new approaches to test and dissect the biophysical properties and the informational content of human mucins. Moreover, the ability to produce large quantities of mucin TR glycodomains provides a substantial first step toward sustainable manufacturing of natural mucin polymers and exploration of their therapeutic opportunities.

## Results

**Engineering strategy for the display of the human mucinome.** Figure 1 presents an overview of the concept for the cell-based display and production of human mucin TR reporters with programmed O-glycan structures. The mucin TR reporters were designed pairwise for either secretion or cell membrane integration through the inclusion of the C-terminal SEA and transmembrane domain of MUC1, and they all included N-terminal GFP, and FLAG tags[15]. We generated a comprehensive set of TR reporters containing approximately 200 amino acids from the TR O-glycodomains of most human secreted and membrane-bound mucins (Fig. 1). The entire sequences selected as representative for each of the human mucin TR O-glycodomains are shown in Supplementary Fig. 1 and Table 1, which also illustrates that mucin TRs are imperfect in sequence but present characteristic patterns of O-glycans. Most of the mucins contain multiple TR sequences and for the mucins MUC2, MUC3, MUC5B, and MUC6 multiple TR reporters were expressed and analyzed.

The transmembrane TR reporters were expressed transiently in glycoengineered HEK293 cells that do not appear to express endogenous mucins, and the secreted reporters were expressed stably[15,26]. We took advantage of our previously reported O-glycoengineering strategy to establish designs for homogeneous

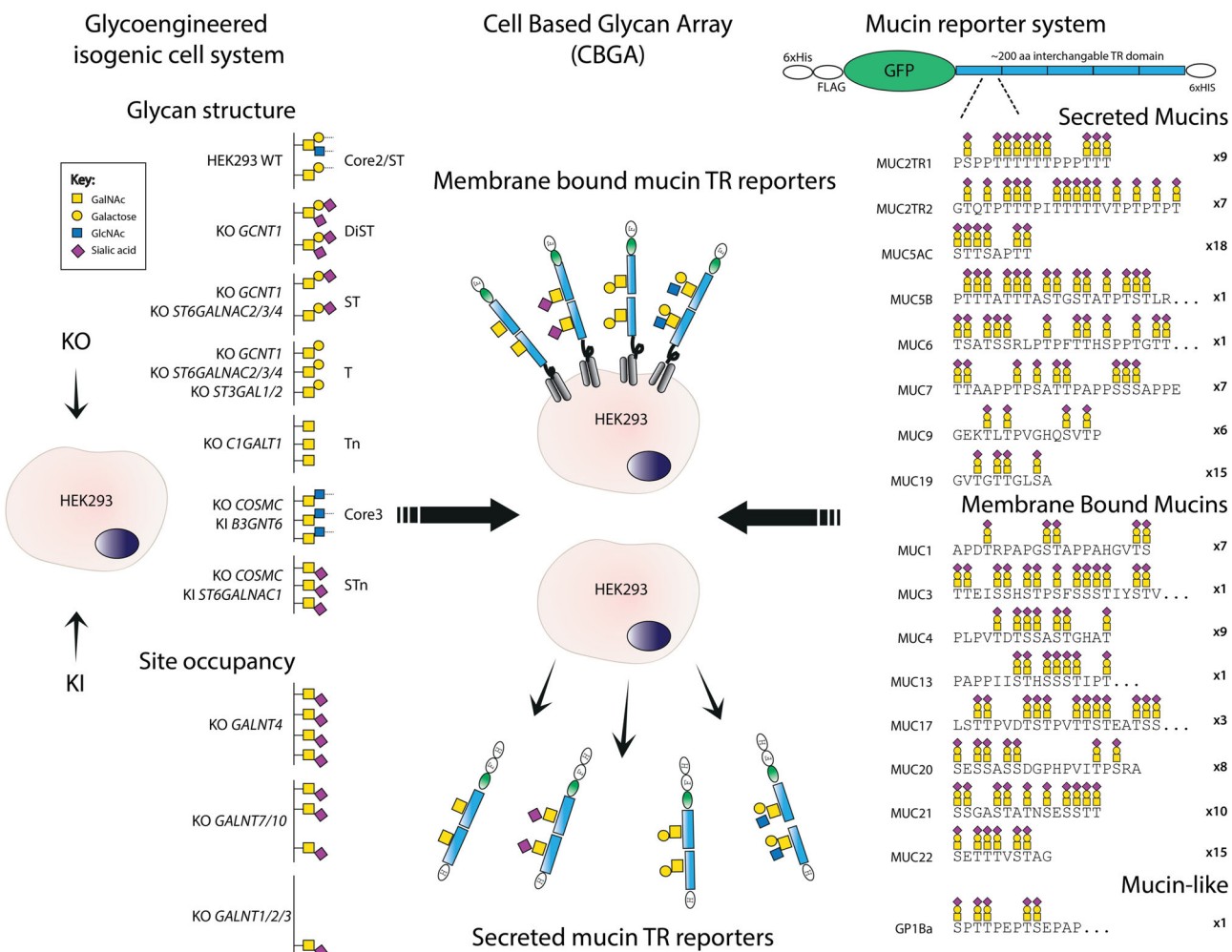

**Fig. 1 Design of the human mucin tandem repeat (TR) display platform.** Illustration of the mucin TR display approach with membrane-bound and secreted mucin reporters expressed in KO/KI glycoengineered isogenic HEK293 cell lines. HEK293 wild-type (WT) cells are predicted to produce a mixture of mSTa, dST, and sialylated core2 structures, and through stable genetic engineering, a library of isogenic HEK293 cells with different O-glycosylation capacities were developed. These cells enable the display of mucin TRs with different O-glycan structures as indicated (glycan symbols and genetic design shown), as well as tunable site occupancy by the engineering of the *GALNT* isoenzyme gene repertoire (left part). The secreted mucin reporter design contains an N-terminal 6xHis and FLAG tag, and GFP followed by different mucin TR domains of ca. 200 amino acids (single TR domains used for MUC3, MUC5B, MUC13, MUC6, and GP1bα), and a second C-terminal 6xHis tag. The membrane-bound mucin reporters contain an N-terminal FLAG tag and GFP followed by the mucin TR domain and further includes the SEA and transmembrane domain of human MUC1 in the C-terminal end for membrane retention. The most characteristic TR sequence for each reporter is illustrated with the number of TRs included (right part). Full sequences of the TRs are shown in Supplementary Fig. 1. Transient or stable expression of the mucin TR reporters in the glycoengineered isogenic HEK293 cell library enables the display of cell surface mucin TRs as well as production of secreted mucin TRs with distinct O-glycan structures. Structures of glycans are shown with symbols drawn according to the Symbol Nomenclature for Glycans (SNFG) format[98].

O-glycosylation capacities that result in the attachment of defined O-glycan structures (Fig. 1). The gene engineering of HEK293 cells included designs for O-glycans designated Tn (KO *C1GALT1*), STn (KO *COSMC*/KI ST6GALNAC1), T (KO *GCNT1*, *ST3GAL1/2*, *ST6GALNAC2/3/4*), monosialyl-T (ST) (KO *GCNT1*, *ST6GAL-NAC2/3/4*), as well as DiST comprised of a mixture of mSTa and disialyl-T (dST) (KO *GCNT1*) (Supplementary Table 2)[17], and the structures, biosynthetic pathways, and genetic regulation are illustrated in Fig. 1. Wild-type HEK293^WT cells produce a mixture of mono and disialylated core1 and core2 structures[15], and KO of *GCNT1* eliminates the core2 structures resulting in a mixture of mono- and disialylated core1 O-glycans (mSTa and dST). KO of *COSMC* or *C1GALT1* results in complete truncation of O-glycans and the uncapped Tn O-glycan without detectable expression of STn[27]. We engineered capacity for core3 (GlcNAcα1-3GalNAcα1-O-Ser/Thr) O-glycosylation by using AAVS1 locus targeted KI of

the core3 synthase (B3GNT6) on top of KO of *COSMC* to eliminate competition from the core1 synthase. To further customize the O-glycan occupancy on mucin TRs, we used KO of *GALNT4* and *GALNT7/10*. The current state of the HEK293 O-glycoengineered cell libraries is listed in Supplementary Table 2.

**Validating the cell-based mucin TR display platform.** We previously verified the general glycosylation outcomes of most of the glycoengineering performed in HEK293 cells[15]. Here, we tested a subset of transiently expressed membrane-bound TR reporters with lectins and monoclonal antibodies (mAbs) with well-characterized specificities for distinct O-glycan structures (Supplementary Fig. 2a, b). There was a substantial window of signal difference in flow cytometry for HEK293 cells with and without expression of the GFP-tagged mucin TR reporter. Thus, the engineered glycosylation capacity for Tn, T, and STn

O-glycosylation could be shown both with the cell population not expressing the mucin TRs (GFP negative) and the transfected cell population expressing these (GFP positive), albeit with higher intensities when mucin TRs were expressed.

We also probed the mucin TR reporters with a panel of mAbs directed to human mucin TR regions, most of which are known to be affected by O-glycosylation either because glycosylation interferes with or blocks binding to the protein core (e.g., mAb to MUC1 such as SM3 or 5E10)[28] or because O-glycans are required for the binding (e.g., mAbs to Tn-MUC1 (5E5), Tn-MUC2 (PMH1), and Tn-MUC4 (3B11))[29–31] (Supplementary Fig. 2c). The observed reactivity patterns were in agreement with the reported specificities of the tested mAbs. We also screened a larger panel of available anti-MUC1 mAbs for reactivity with different glycoforms of MUC1, which confirmed specific reactivities with defined glycoforms (Tn-MUC1 for 5E5 and 2D9; T-MUC1 for 1B9)[29], while the binding of antibodies to the PDTR peptide region (SM3, HMGF1, and 5E10) revealed different glycoform selectivities[32] (Supplementary Fig. 2d).

**Structural analysis of secreted mucin TRs.** Secreted TR reporters stably expressed in glycoengineered HEK293 cells were isolated by Ni-chromatography and assessed by SDS-PAGE analysis, which showed that the GFP-tagged proteins migrated as distinct rather homogeneous bands (Supplementary Figs. 3 and 4). We used LysC digestion to liberate the intact TR O-glycodomains and C4 and C18-HPLC to purify these for further analysis (Supplementary Fig. 5). For direct intact mass analysis of mucin TRs, we used pretreatment with neuraminidase to reduce complexity and facilitate deconvolution and interpretation.

The dense O-glycosylation of mucin TRs in most cases blocks cleavage by peptidases limiting conventional glycoproteomics strategies[24,33]. However, the MUC1 TRs are cleavable by endoproteinase-Asp-N (AspN) in the PDTR sequence[34–36], and we, therefore, used the MUC1 reporter for full characterization (Fig. 2). The MUC1 reporter contains 34 predicted O-glycosites and includes six 20-mer TRs and a C-terminal TR, where the last GVTSA sequence proceeds into the 6xHis tag. We used LysC to cleave the purified GFP-tagged reporter and isolated the TR O-glycodomain for LC–MS intact MS analysis (Fig. 2a). The simplest Tn glycoform (HEK293$^{KO\ C1GALT1}$) revealed a rather small range of incremental masses corresponding to HexNAc (203.08) centered around the predicted protein size (m/z 14,902.14) with 28–35 HexNAc residues, while the T (HEK293$^{KO\ GCNT1,\ ST3GAL1/2,ST6GALNAC2/3/4}$) and ST (HEK293$^{KO\ GCNT1}$) glycoforms after neuraminidase treatment generated the same narrow range of predicted 28–35 Hex-HexNAc disaccharides. In contrast, the STn glycoform (HEK293$^{KO\ C1GALT1\ KI\ ST6GALNAC1}$) analyzed after treatment with neuraminidase produced a slightly broader range of detectable glycoforms from 18–35 HexNAcs, suggesting that ST6GALNAC1 competes partly with the completion of GalNAc glycosylation by GALNTs, in agreement with previous studies[37,38]. Analysis of the MUC1 TR reporters after AspN digestion revealed that the predominant 20-mer glycopeptides derived from Tn-MUC1 and STn-MUC1 were those with 4–5 O-glycans per TR (Fig. 2b). For the bottom-up analysis, we were required to use pretreatment with neuraminidase because the sialylated glycoforms were poorly digested by AspN. For the MUC1 reporters with higher glycan complexity (T-MUC1 and ST-MUC1) the most abundant glycopeptide variants appeared to be shifted toward 3–4 O-glycans per TR (Fig. 2b), however, this result may be biased by inefficient AspN digestion since the intact MS analysis did not show the same tendency (Fig. 2a).

As an illustrative example for the ability to engineer the occupancy of O-glycans in mucin TRs, we analyzed the MUC1 TR reporter produced in cells with and without KO of GALNT4 (HEK293$^{KO\ COSMC,\ GALNT4}$) (Supplementary Fig. 6). KO of GALNT4 resulted in reduced HexNAc incorporation with the major proteoform centered around 28 HexNAc residues indicating loss of 6–7 O-glycans, which is in agreement with previous in vitro studies and predicted to represent the loss of glycosylation in the PDTR sequence motifs[39]. Finally, we confirmed the glycoengineering by O-glycan profiling of released O-glycans from the MUC1 TR reporters with four different O-glycan designs by MALDI-TOF analysis (Supplementary Fig. 7).

The promising results obtained with intact MS analysis of the MUC1 TR glycodomains prompted intact MS analysis of the simplest Tn glycoforms of MUC2, MUC5AC, MUC7, MUC13, and MUC22 TR glycodomains, which showed similarly high occupancy of available glycosites with rather homogeneous patterns (Fig. 3). For most mucin TRs the proteoform with the highest number of HexNAc residues correlated with the number of potential O-glycosites with the most abundant proteofoms centered close to or a little lower than this. However, for the MUC22 TR reporter the highest abundant peaks were centered around 68–71 with 83 potential O-glycosites suggesting lower occupancy.

Interestingly, we noted minor molecular species with an apparent excess of one HexNAc compared to the total number of Ser/Thr potential O-glycosites, and we anticipate that this is due to a very low degree of HexNAc-HexNAc disaccharide incorporation. This may provide a basis for the existence of GalNAc-GalNAc O-glycans as reported in human meconium[40]. Similarly, we observed the appearance of low amounts of fucose on mucin TRs expressed in HEK293 cells with KO of GCNT1 and ST3GAL1/2, presumably related to minor capping with blood group H (Supplementary Fig. 7).

**Dissection of the binding specificities of Streptococcal Siglec-like adhesins.** We previously used the cell-based glycan array platform to dissect the binding specificities of two Siglec-like binding regions (BRs) of serine-rich repeat adhesins expressed by oral streptococci, and demonstrated distinct binding specificities of Streptococcus gordonii (Hsa$_{BR}$) for mSTa and Streptococcus mitis (10712$_{BR}$) for disialylated core2 O-glycans displayed on a reporter containing the mucin-like domain of GP1bα (Fig. 1)[15,41,42]. We also found evidence that these adhesins showed binding selectivity for mucin TRs expressed in HEK293$^{WT}$, and to explore these findings further we here included binding studies with the preferred O-glycan structures on different mucin TRs (Fig. 4a). We compared the expression of TRs with core2 O-glycans (HEK293$^{WT}$) and mSTa O-glycans (HEK293$^{KO\ GCNT1,\ ST6GAL-NAC2/3/4}$), which replicated highly select mucin TR binding specificities, and also surprisingly uncovered that O-glycosylation of some mucin TRs do not follow the general glycosylation capacities. Importantly, membrane expression levels were comparable between mucin TR reporters and between isogenic cell lines as confirmed by anti-FLAG staining (Supplementary Fig. 8a). Thus, we found that the selective binding of Hsa$_{BR}$ to MUC2#1, MUC7, and GP1bα was unaffected by limiting O-glycans to mSTa (KO ST6GALNACs), while binding, especially to GP1bα, was markedly increased. Moreover, the binding of 10712$_{BR}$ to GP1bα was abolished by limiting O-glycans to mSTa since this removes the preferred core2 O-glycan structures. This suggests that the mucin TR sequences to some extent affect the structural outcome of O-glycosylation with respect to modifications of the innermost GalNAc residue, i.e., core2 branching by addition of β1-6GlcNAc (directed by GCNT1) and/or sialylation by addition of sialic acid (α2-6Neu5Ac) (directed by ST6GALNACs). The most remarkable finding was that the MUC2 TRs (especially the MUC2 region

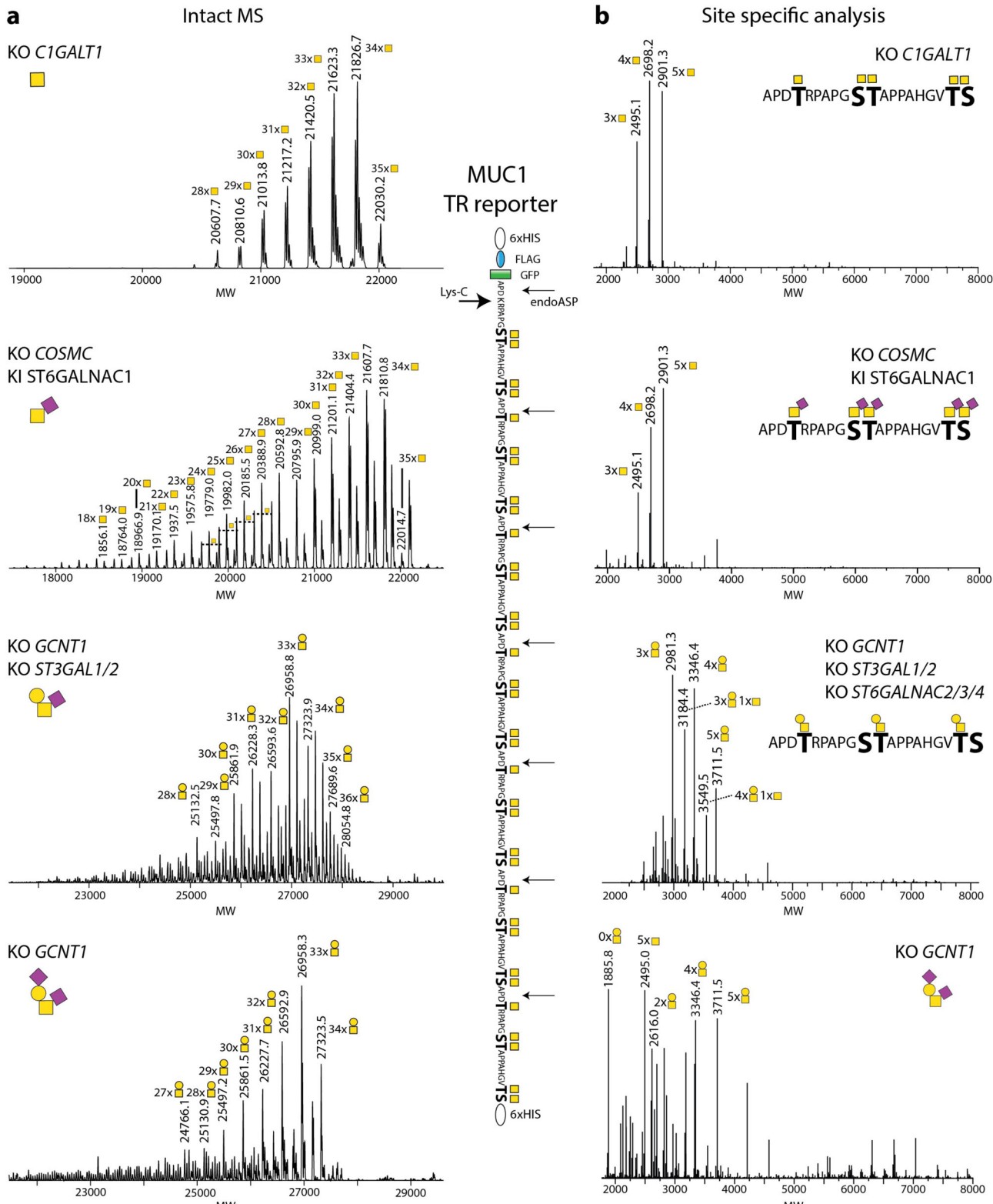

**Fig. 2 Mass spectrometry analysis of secreted MUC1 TR reporter O-glycoforms. a** Deconvoluted intact mass spectra of secreted, purified MUC1 reporter produced in HEK293^KO C1GALT1, HEK293^KO COSMC, KI ST6GALNAC1, HEK293^KO GCNT1, ST3GAL1/2, and HEK293^KO GCNT1 cells. Reporters were treated with neuraminidase to remove sialic acids and reduce complexity, and digested by Lys-C followed by HPLC C4 isolation yielding the 157 amino acid MUC1 TR O-glycodomain fragment. All MUC1 O-glycoforms (Tn and T) showed a rather homogeneous mass comprising 32–35 HexNAc/HexHexNAc residues, and with 33 or 34 HexNAc/HexHexNAc being the most abundant peak. For all intact mass spectra experimentally determined and theoretically calculated masses were composed in Supplementary Table 4. **b** Site-specific O-glycopeptide LC–MS/MS analysis of MUC1 reporters after AspN digestion.

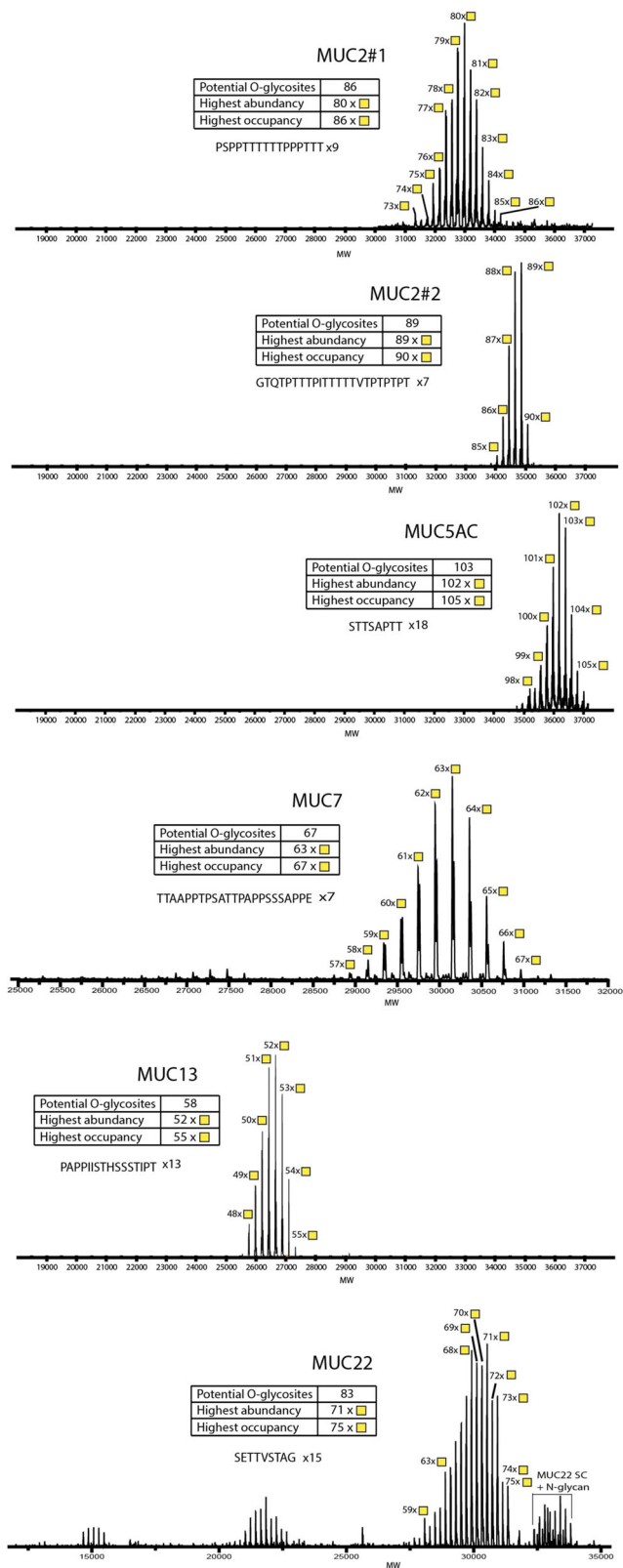

**Fig. 3 Intact mass spectrometry analysis of isolated O-glycodomains of mucin TR reporters.** Deconvoluted intact mass spectra of secreted, purified mucin TR reporters expressed in HEK293$^{KO\ C1GALT1}$. The mucin TR reporters designed from secreted mucins (MUC2TR#1, MUC2TR#2, MUC5AC, and MUC7) and membrane-bound classified mucins (MUC13 and MUC22) were analyzed, the most abundant masses are annotated with the predicted number of attached HexNAc residues. For all intact mass spectra experimentally determined and theoretically calculated masses were composed in Supplementary Table 5.

TRs (Fig. 4a). Similar to Hsa$_{BR}$ this binding was increased when these reporters were expressed in HEK293$^{KO\ GCNT1,\ ST6GALNAC2/3/4}$ with homogenous mSTa O-glycosylation. Collectively, our results support the notion that the Siglec-like adhesins recognize specific O-glycans, and that this recognition is co-determined by the context provided by the mucin TR sequence. While further studies are still needed, it is likely that recognition of clustered patches or patterns of multiple O-glycans is involved[43,44]. Simple multivalent presentation of O-glycans does not seem to be a major determining factor since all the TR reporters studied are predicted to present multiple closely spaced O-glycans. Mucin TRs and mucin-like domains such as the stem region of GP1bα are characterized by dense O-glycosylation, and the positions and patterns of the O-glycan decoration is determined by the peptide sequence (distribution of Ser/Thr residues) and the specificities of the available polypeptide GalNAc-transferases (GALNT1-20) that control the initiation of O-glycosylation[45]. Analysis of the amino acid sequences used for the TR reporters derived from human mucins and GP1bα did not reveal simple common sequence motifs shared among those providing binding for the Siglec-like adhesins, and thus, the data do not allow us to define the recognition motifs in further detail. Recognition of clustered saccharide patches orchestrated by positions, spacing, and direct interactions of multiple glycans in a protein was earlier proposed to provide expanded binding specificities and high-affinity interactions, and evidence in support of this has been found with all types of glycoconjugates[43,44].

**Characterization of the mucin-degrading activity of the glycoprotease StcE.** Secreted protease of C1 esterase inhibitor (StcE) from Enterohemorrhagic *Escherichia coli* (EHEC) O157:H7 is a zinc metalloprotease with a remarkable ability for cleaving densely O-glycosylated mucins and mucin-like glycoproteins. StcE is thought to serve in colonic mucin degradation facilitating EHEC adherence to the epithelium and subsequent infection[46,47]. Recent studies demonstrated that StcE has selective substrate specificity for S/T-X-S/T motifs with a requirement for O-glycans at the first S/T residue[48]. Moreover, a catalytically inactive mutant of StcE (StcE$^{E447D}$) has glycan-binding properties for the dST core1 O-glycan as evaluated by printed glycan arrays[49], but also exhibits general binding properties for mucins[48,50–52]. We used our purified TR reporters and those displayed on cells to further explore the fine substrate specificity of the recombinantly purified StcE glycoprotease and to dissect its reported mucin-binding properties (Fig. 5). First, we found that StcE efficiently cleaved isolated MUC2 and MUC5AC reporters with core2 (HEK293$^{WT}$) and Tn O-glycans already at low concentrations of 10-40 ng/mL (1:2500–500, StcE: TR reporter ratio), while the STn glycoform was essentially resistant to cleavage (Fig. 5b and Supplementary Fig. 9a). Next, we explored the selectivity of StcE with the 20 membrane-bound mucin TRs displayed on HEK293$^{WT}$ cells by monitoring loss of the N-terminal FLAG tag by flow cytometry using fluorescent anti-FLAG tag antibodies (Fig. 5a). StcE efficiently cleaved most of the mucin TRs with the notable exception of MUC1 and MUC20, as well as the control TR reporter

designated #1) had a relatively high proportion of core1 O-glycans with only α2-3 linked sialic acids (mSTa) installed in HEK293$^{WT}$ cells, while most other mucin TRs displayed a mixture of core2 and dST (Fig. 4b).

We also tested another Siglec-like BR, *Streptococcus gordonii* GspB$_{BR}$, which like Hsa$_{BR}$ binds to ST, but showed a very different binding pattern with the highest binding to MUC7 and MUC22

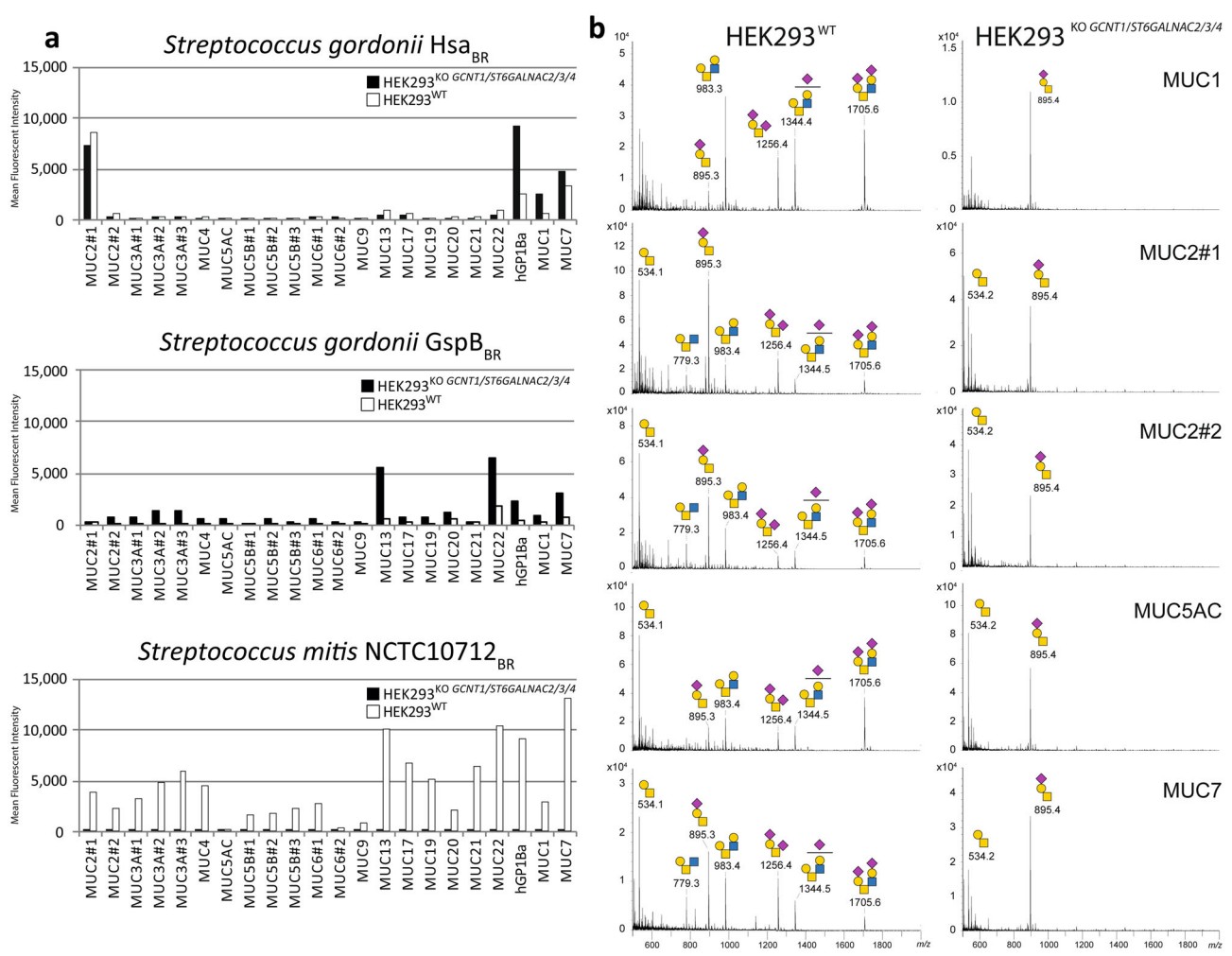

**Fig. 4 The cell-based mucin display reveals binding specificities of streptococcal Siglec-like adhesins. a** Flow cytometry analysis of Siglec-like adhesin BRs binding to mucin TR reporters expressed transiently in HEK293[WT] and HEK293[KO GCNT1, ST6GALNAC2/3/4] cells. Bar diagrams show mean fluorescence intensities (MFI). *S. gordonii* adhesins (Hsa[BR] and GspB[BR]) exhibit binding to mSTa O-glycans with different preferences for mucin TRs, while the *S. mitis* adhesin (NCTC10712[BR]) exhibits binding to disialylated core2 O-glycans without a strong preference for mucin TRs. Representative data of three independent experiments are shown with subtracted binding levels to cells without expression of the mucin TR reporters using GFP expression to qualify positive/negative mucin TR expression. The data without subtraction is shown in Supplementary Fig. 8b. **b** MS spectra of O-glycoprofiling of purified mucin TRs produced in HEK293[WT] cells illustrating relatively low levels of core2 O-glycans on the MUC2#1/2 TRs. Mucin TR reporters produced in HEK293[KO GCNT1, ST6GALNAC2/3/4] cells in contrast show similar T/mSTa O-glycan profiles. Predicted O-glycan structures are shown based on compositional analysis and the genetic engineering design. Source data are provided as a Source Data file.

designed with a single O-glycosite (Fig. 5c and Supplementary Fig. 9c, d). Dose-titration analysis in both assays showed low ng/mL cleavage for most mucin TR reporters, while no cleavage of the MUC1 TR reporter was found even at 10 μg/mL (Supplementary Fig. 9b, d). StcE was shown previously to cleave the entire MUC1 expressed on cancer cells, but this may be due to cleavage outside the TR region as the proposed StcE cleavage motif (S/T-X-S/T) is absent from the well-conserved TRs[48,52]. Finally, we dissected the effect of O-glycan structures on StcE cleavage using the MUC2 and MUC5AC TR reporters, and found that core1 and core2 O-glycans including Tn are efficiently cleaved, while STn, as well as core3 O-glycans, efficiently blocked proteolysis (Fig. 5d and Supplementary Fig. 9e).

**Characterization of the mucin-binding properties of StcE.** Next, we explored the intriguing suggestion that StcE plays a role in adherence of EHEC to the intestinal epithelium by binding mucins[47,49,53]. A catalytically inactive mutant of StcE (StcE[E447D])

was recently shown to exhibit broad binding to mucins in tissue sections and suggested to recognize isolated dST O-glycans on a glycan array[52]. Examination of the 3-D structure of StcE[49] revealed that the protein contained a C-terminal domain (here designated X409) opposite to the catalytic metalloprotease domain (M66), which we predicted could represent an evolutionarily mobile binding module. In line with previous findings, deletion of this X409 module did not affect the catalytic activity of StcE with the WT MUC2 reporter (Fig. 6a, b and Supplementary Fig. 10a)[49], and the X409 module alone did not exhibit enzymatic activity (Fig. 6b). However, deletion of the X409 module completely abrogated StcE binding to mucin-producing cells in human colonic tissues, whereas the X409 alone (GFP-tagged) reproduced the strong tissue staining properties of the intact StcE[WT] (Fig. 6c). Note also that the binding of active StcE, inactive StcE[E447D], as well as X409 alone fully recapitulated the tissue binding found previously with the inactive StcE[E447D] mutant[48,50–52]. Interestingly, the StcE/X409 staining appeared to overlap with MUC2 expression in the human colon as shown by

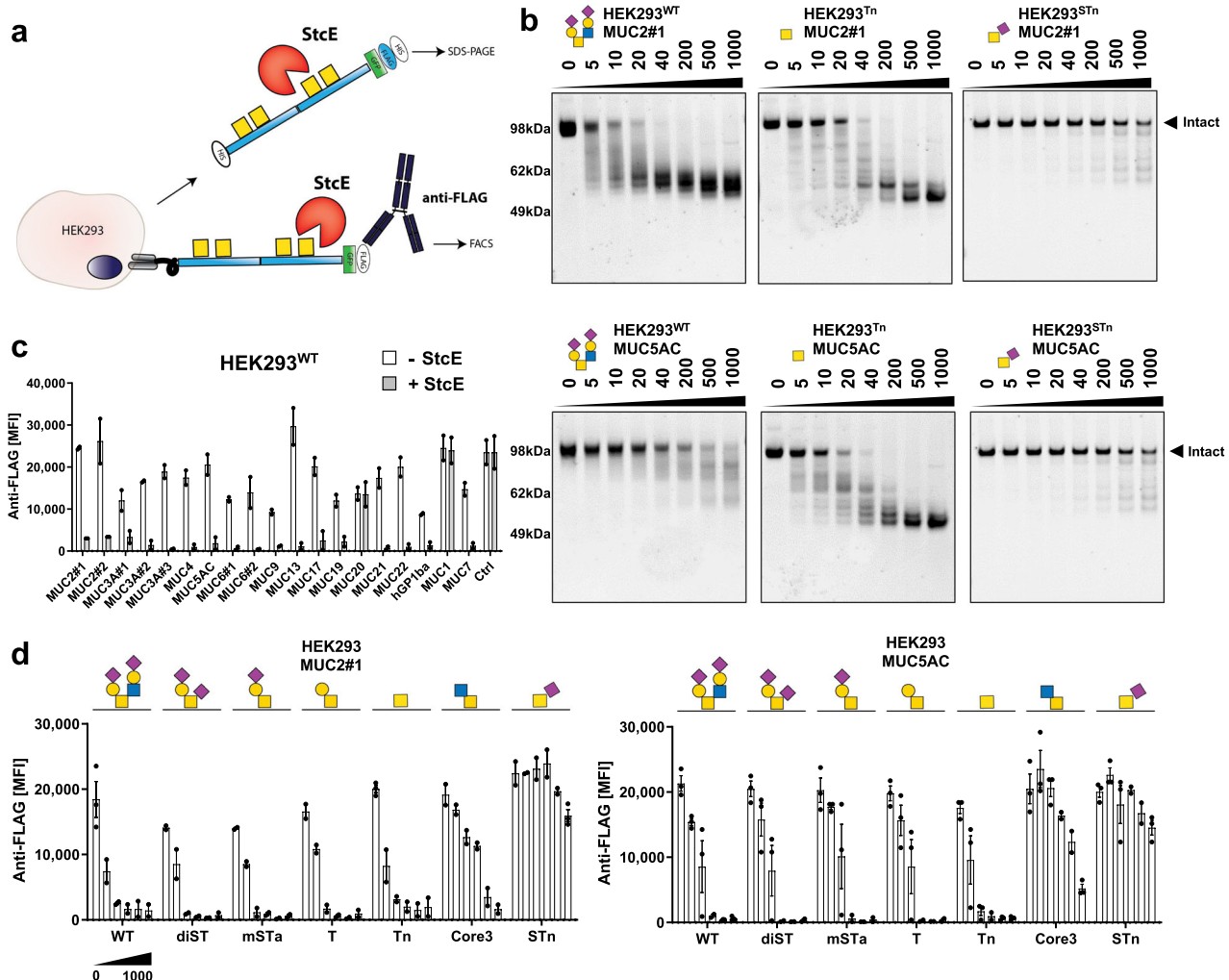

**Fig. 5 The glycoprotease StcE cleaves selective mucin TRs and O-glycoforms. a** Graphic depiction of the assay designs for enzyme assays with isolated and cell membrane-bound mucin TR reporters. **b** SDS-PAGE analysis of StcE digestion (dose titration) of isolated MUC2#1 and MUC5AC reporters produced in glycoengineered HEK293 cells (WT, Tn, and STn O-glycoforms). Purified reporters (0.5 µg) were incubated for 2 h at 37 °C, and gels visualized with Krypton fluorescent protein stain. Representative gels of three independent experiments are shown. **c** Flow cytometry analysis of StcE cleavage of different TR reporters expressed stably as membrane-bound proteins on the surface of HEK293^WT cells and detected by anti-FLAG antibody binding. Bar diagram shows average mean fluorescent intensities (MFI) ± SEM. An artificially designed TR reporter with a single O-glycosite was used to serve as a control (ctrl) for the clusters and patterns of O-glycans found in human mucin TRs. Data from three independent experiments are shown. **d** Flow cytometry analysis of StcE cleavage (0–1000 ng/mL dose titration) of MUC2#1 and MUC5AC membrane reporters stably expressed on glycoengineered HEK293 cells (core2, dST, mSTa, T, Tn, core3, and STn glycoforms). Note that core3 is represented as the core disaccharide (GlcNAcα1-3GalNAcα1-O-Ser/Thr), but this is likely galactosylated and sialylated. Data are presented as average MFI ± SEM of three independent experiments. Source data are provided as a Source Data file.

co-localization of staining with a mAb directed against Tn-MUC2 (PMH1) (Supplementary Fig. 10b)[30]. The X409 module displayed strong binding to normal human colon and stomach tissues and colon and stomach cancers. While we found low binding to other normal tissues including the pancreas and breast, strong binding to the counterpart cancer tissues was observed (Supplementary Fig. 10c). Probing the X409 module with the cell-based mucin TR display revealed that this binding module bound to select human mucin TRs (carrying core2 O-glycans), and e.g., not to MUC1 (Fig. 6d). Moreover, the strong binding to MUC2 and MUC5AC was only slightly influenced by the O-glycan structures attached to the TRs, although weaker binding to TRs carrying Tn, core3, and especially STn O-glycans was observed (Fig. 6e). These results clearly demonstrate that the X409 module mediates the mucin-binding properties of StcE, while the catalytic unit of StcE may mediate selective binding to dST O-glycans[49]. Here, we did

not pursue this glycan-binding property of the catalytic unit of StcE given the focus on the mucin TRs, but a recent preliminary report describing the use of the full StcE^E447D mutant for affinity isolation of O-glycoproteins, suggests that the enzyme enriches not only mucins but also a variety of other O-glycoproteins[54]. Given that StcE, StcE^E447D, and X409 alone exhibited the same highly selective tissue binding to mucin-producing cells, it is likely that the dST glycan-binding properties of the catalytic unit of StcE found with a high density of glycans printed on glass slides[49], does not contribute substantially to binding to tissues. In conclusion, our results show that X409 does not directly impact the catalytic activity of StcE, and it is likely that this domain serves to target StcE to mucins through its highly selective mucin-binding properties[47,49]. The X409 module may be exploited to probe for the expression of mucins in normal and cancer tissues.

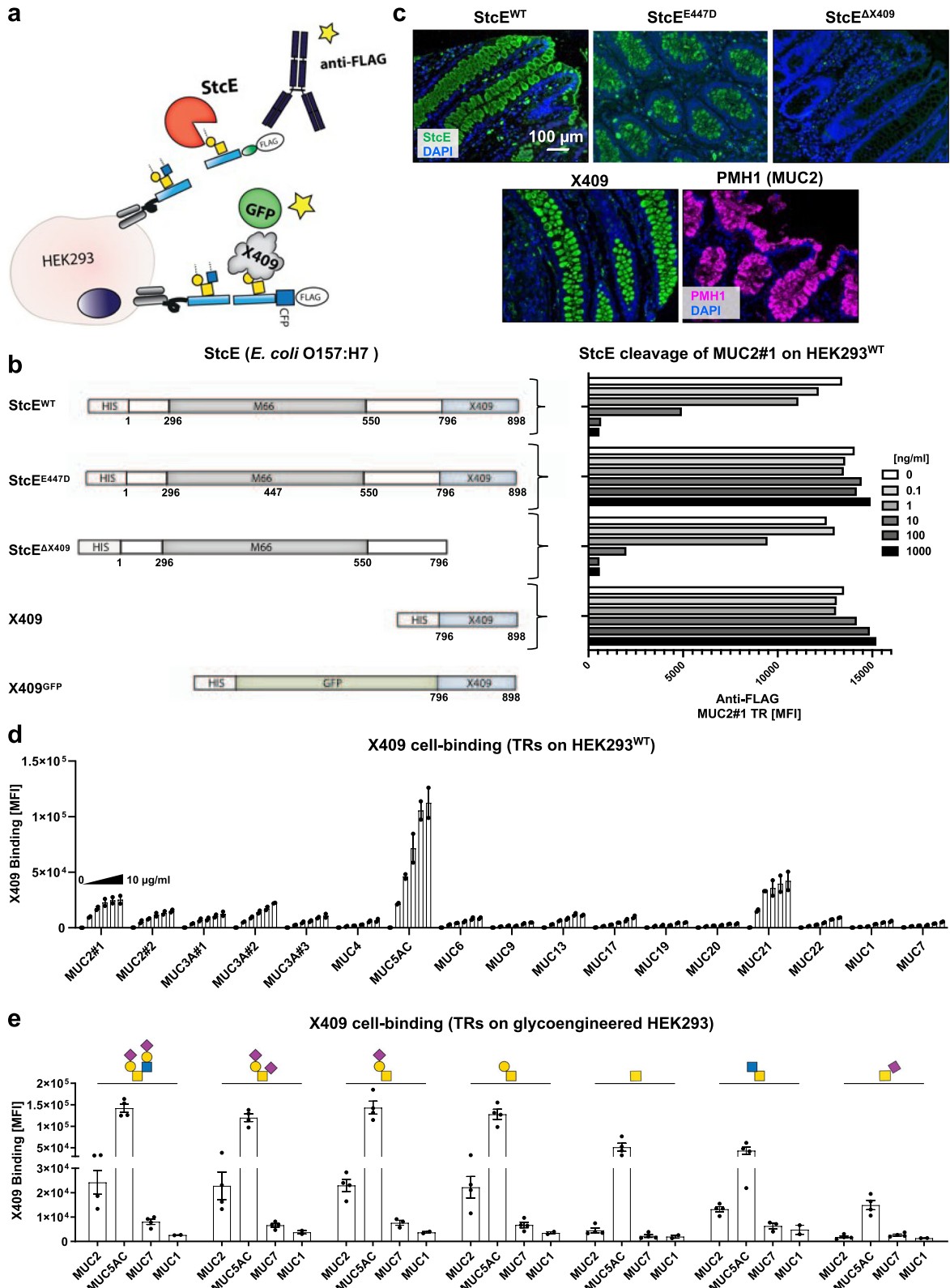

**Analysis of influenza virus receptor interaction**. Influenza A virus (IAV) employs a trimeric hemagglutinin (HA) that recognizes sialoglycans to bind and infect host cells, and a neuraminidase (NA) to cleave sialic acid and destroy the HA receptors for virus release. IAV interacts with mucins lining the intranasal, tracheal, and lung surfaces, and HA binding and NA cleavage of mucins have been suggested to play an important role in the penetration of the mucus layer and infection of respiratory tissues[55]. Here, we addressed if the sequence and glycosylation of four mucin TRs produced in HEK293^WT or HEK293^KO GCNT1 influenced IAV HA-NA activity by analyzing binding and dissociation kinetics of the laboratory mouse-adapted influenza strain A/Puerto Rico/8/1934 (H1N1) (PR8) virions (α2-3Neu5Ac linkage specificity) to loaded sensors using biolayer interferometry[56,57].

**Fig. 6 The X409 domain mediates the mucin binding of StcE. a** Graphic depiction of the design of enzyme activity and binding assays performed. **b** Schematic representation of expression construct designs for full coding StcE$^{WT}$, StcE$^{E447D}$, X409 domain truncated StcE (StcE$^{\Delta X409}$), and the isolated X409 domains ± GFP (left). Flow cytometry analysis of cleavage (dose titration) of membrane-bound MUC2#1 expressed in HEK293$^{WT}$ cells by StcE$^{WT}$, StcE$^{E447D}$, StcE$^{\Delta X409}$, and the X409 domain detected by anti-FLAG staining and flow cytometry (right). Representative data of three independent experiments with similar results are shown. **c** Immunofluorescence staining of representative sections of normal colon stained with StcE, StcE$^{E447D}$, StcE$^{\Delta X409}$, X409, and anti-MUC2 mAb (PMH1). Counterstained with DAPI (blue). Scale bars = 100 μm. **d** Flow cytometry analysis of X409-GFP binding (0–10 μg/mL dose titration) to HEK293$^{WT}$ cells stably expressing mucin TR reporters tagged with CFP (instead of GFP). Average MFI ± SEM values from two independent experiments are shown. **e** Flow cytometry analysis of X409-GFP binding (1 μg/mL) to MUC2#1, MUC5AC, MUC7, and MUC1 reporters expressed on glycoengineered HEK293 cells (core2, dST, mSTa, T, Tn, core3, and STn). Data are presented as average MFI ± SEM of two (MUC1) or four (other MUCs) independent experiments. Source data are provided as a Source Data file.

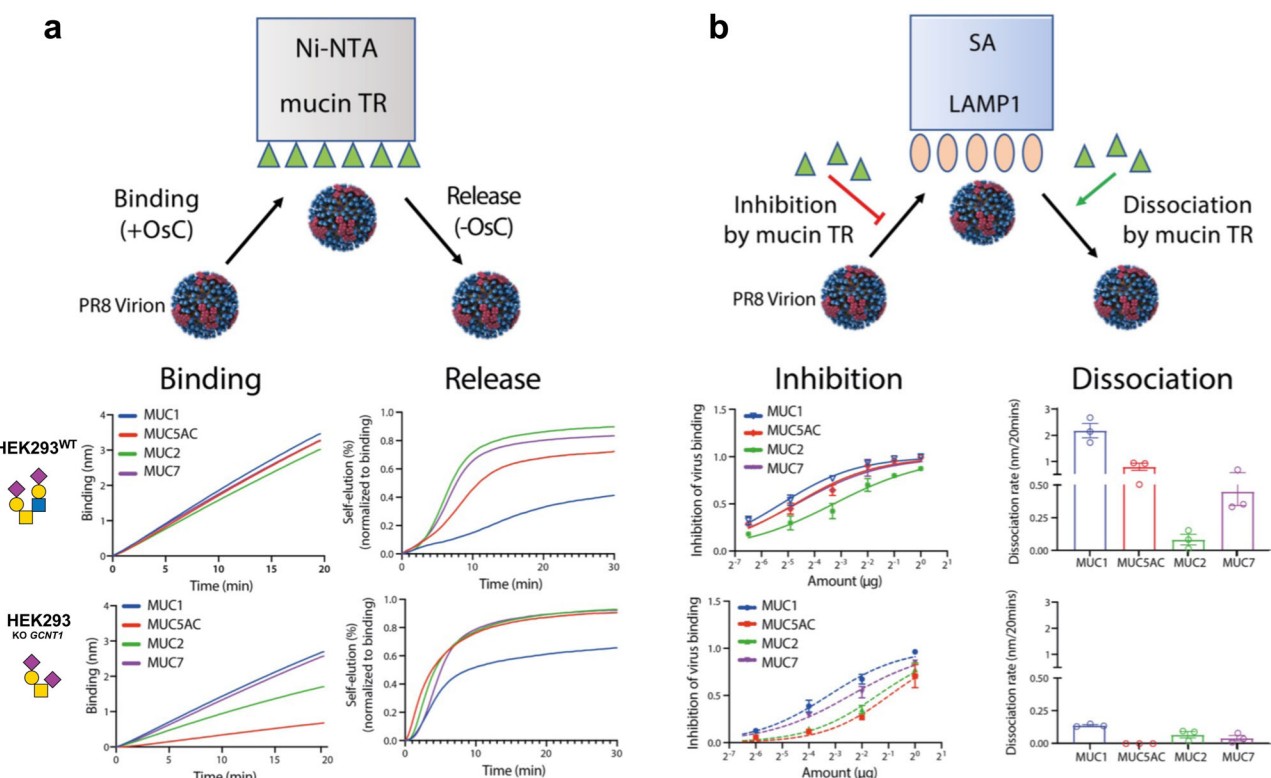

**Fig. 7 Selective interactions of Influenza A virus (IAV) with mucin TR reporters. a** Cartoon (top) illustrates assay design for IAV binding dynamics to sensor-immobilized mucin TR reporters (HA-mediated binding and NA-mediated release). Graphs show binding of PR8 IAV to probes loaded with MUC1, MUC2, MUC5AC, or MUC7 TR reporters produced in HEK293$^{WT}$ or HEK293$^{KO\ GCNT1}$ in the presence of NA inhibitor (OsC), and self-elution of PR8 IAV from mucin TR reporters after removal of OsC is shown in time. **b** Cartoon (top) illustrates assay design for competition dynamics of soluble mucin TR reporters with IAV binding to sensor-immobilized LAMP1 (inhibition and dissociation) using biolayer interferometry analysis. Graphs show inhibition of PR8 binding to LAMP1-loaded probes by increasing concentrations of soluble mucin TR reporters produced in HEK293$^{WT}$ or HEK293$^{KO\ GCNT1}$ as well as the ability of soluble mucin TR reporters to dissociate PR8 bound to LAMP1 from the probe. All graphs show representative data of three independent experiments as average values ± SEM. Source data are provided as a Source Data file.

Biolayer interferometry enables the interactional study of the PR8 virions with glycans presented in their native protein-linked form. In the presence of the NA inhibitor OsC[58], virus particles bound to sensors saturated with MUC2, MUC5AC, MUC7, and MUC1 TRs carrying core2 O-glycans (produced in HEK293$^{WT}$) with similar kinetics (Fig. 7a). This binding was markedly slowed with mucin TRs carrying core1 O-glycans only (produced in HEK293$^{KO\ GCNT1}$), with the largest reduction observed for MUC5AC. NA activity-dependent self-elution of virions bound to similar levels to the different mucin TR coated sensors, was monitored after removal of the OsC NA inhibitor (Fig. 7a). This self-elution reflects the HA-NA-receptor balance as it depends on HA avidity and NA activity for the receptors present on the sensor surface and on the receptor density[55]. Self-elution was slowest from MUC1-coated sensors, remarkably despite the fact that the

MUC1 TR reporter contains fewer O-glycans compared to the other TR reporters. Self-elution from mucin TRs with core1 O-glycans was faster compared to their counterparts with core2 O-glycans, and still the slowest release was observed with MUC1 TR reporter (Fig. 7a). Next, the different mucin TR reporters were assessed for their ability to compete with PR8 virion binding to sensors coated with recombinant soluble lysosomal-associated membrane glycoprotein 1 (LAMP1), which carries multiple N-glycans with α2-3 and α2-6 linked sialic acids and serves as a potent IAV receptor[59]. Overall, mucin TR reporters with core2 O-glycans displayed stronger competition with LAMP1 binding than TRs carrying only core1 O-glycans (Fig. 7b). The MUC1 TR with core2 O-glycans was most effective and the MUC5AC with only core1 O-glycans was least effective in competing with LAMP1 (Fig. 7b). Similar results were obtained by mucin-driven elution

(in the absence of NA activity by the presence of OsC) of virus bound to LAMP1-loaded sensors, demonstrating that mucin TR reporters can efficiently compete with PR8-LAMP1 interactions, depending on their specific structure. Collectively, these results indicate that IAV particles exhibit preferential binding to mucin TRs and core2 O-glycans.

## Discussion

The cell-based mucin display platform presented here offers a unique resource with wide applications and opportunities for the discovery and dissection of molecular properties of natural human mucins and other glycoproteins with mucin domains. The informational cues harbored in mucin TRs with their distinct patterns and structures of O-glycans can now be addressed with well-defined molecules in a variety of assay formats. This was illustrated by analyzing the binding properties of streptococcal Siglec-like adhesins revealing requirements for specific O-glycan structures and select mucin TR sequences (Fig. 4). We used the mucin display to dissect the fine substrate specificity of the mucin-destroying glycoprotease StcE derived from pathogenic EHEC[46,47], demonstrating clear selectivity for both distinct mucin TRs and O-glycoforms, and importantly discovering that the normal core3 O-glycosylation pathway in the colon actually inhibits StcE digestion of MUC2 (Figs. 5 and 6). Studies with IAV also suggest that binding to and release from mucins are not simply determined by the number of O-glycans and sialic acids, but partly driven by cues encoded by the TR sequences and O-glycan structure (Fig. 7). The mucin TR reporter production platform also provided unprecedented insights into the O-glycosylation of mucins and enabled intact MS analysis to demonstrate surprising efficiency in the initiation of O-glycosylation by the polypeptide GalNAc-Ts (Figs. 2 and 3).

Mucins arguably represent the last frontier in the analytics of glycoproteins. Most mucins are extremely large and heterogeneous glycoproteins that are resistant to conventional glycoproteomics strategies dependent on proteolytic fragmentation and sequencing[24,25,33], and despite increasing knowledge of O-glycosites[20], identification of actual sites of glycosylation in mucins is essentially limited to MUC1[34–36], lubricin[25], and the large N-terminal mucin-like region of MUC16[20]. The platform developed here for the production of secreted mucin TR reporters enabled us to produce representative fragments of the large mucin TR regions and to analyze these by intact MS analysis, which demonstrated that the TRs of the mucins tested are O-glycosylated to near completion at all putative Ser/Thr glycosites with high fidelity (Figs. 2 and 3). This analysis was only performed with the simplest Tn O-glycosylation, which may be biased toward higher efficiency in the incorporation of GalNAc residues by the many GALNTs in the absence of competition from elongation[60,61]. However, using MUC1 as a model it was possible to analyze the more complex ST and STn glycoforms, after removal of sialic acids, to demonstrate that elongation by the core1 synthase (*C1GALT1*) in this cell model did not significantly interfere with O-glycan occupancy, while α2-6 sialylation by *ST6GALNAC1* did appear to slightly interfere as observed by an increase in proteoforms with lower occupancy (Fig. 2). Overexpression of ST6GALNAC1 in cell lines overrides the O-glycan elongation process leading to accumulation of the cancer-associated STn O-glycan[37,38]. We engineered homogenous expression of STn in HEK293 cells by eliminating O-glycan elongation (KO *COSMC*) and by introducing ST6GALNAC1 using site-specific KI, which does not lead to substantial overexpression[15]. We can therefore not assess the effects of expression of *ST6GALNAC1* on O-glycan elongation, but it is interesting to note that even without overexpression the efficiency

of O-glycosylation initiation was slightly affected (Fig. 2a). ST6GALNAC1 may compete with the initiation of O-glycosylation orchestrated by the many GALNTs that utilize C-terminal lectin modules for binding to GalNAc residues and efficient incorporation of GalNAc at all glycosites. The STn O-glycan is generally not expressed in normal cells[62–64], but interestingly high expression of 9-O-Acetylated STn is selectively found in the normal colon[63,65], showing that the capacity for STn O-glycosylation is present and may compete with the normal core3 pathway directed by B3GNT6. Further expansion of the cell-based glycan array through the engineering of the sialic acid acetyltransferase will allow us to address 9-O-acetylation of STn.

For a long time mucins have represented a black box in exploring the molecular cues that serve in intrinsic interactions with glycan-binding proteins and in extrinsic interactions with microorganisms[66]. Dissection of interactions with simple O-glycan structures found on mucins have benefited tremendously from the development of printed glycan arrays[67,68], and these have for decades served as essential tools in exploring the interactome of glycans and proteins[69]. However, mucins and their large variable TR domains present O-glycans in different densities, and patterns which are likely to provide more specific interactions and instructive cues. Mucin TRs differ markedly in sequence, length, and numbers within closely related mammals[70], and this divergence in TRs may have evolved to accommodate specific recognition of higher-order patterns and clusters of O-glycans[15]. We previously provided evidence for this by use of the cell-based glycan array demonstrating that two distinct streptococcal Siglec-like adhesins bind selectively to O-glycans presented on distinct mucin-like domains in O-glycoproteins and mucins[15]. This prompted us to develop the cell-based mucin display combining TR reporters with glycoengineered HEK293 isogenic cells (Fig. 1), which enabled us to demonstrate that three adhesins (Hsa$_{BR}$, GSP$_{BR}$, and NCT10712$_{BR}$) bind to different O-glycan structures and to distinct subsets of mucin TRs (Fig.4a). Moreover, the select binding of the Hsa$_{BR}$ adhesin to O-glycans on MUC2 could be installed in the NCT10712$_{BR}$ adhesin by mutating a loop found in Hsa$_{BR}$[71]. The distinct binding specificities of these adhesins have an impact on their role in endocardial infection and adherence to platelets and aortic valves[41]. More recently, the mucin display was also used to demonstrated that several Siglecs selectively recognize sialylated O-glycans in the context of select mucin TRs[72]. The mucin display platform can clearly be used to discover and dissect interactions with clusters and patterns of O-glycans on mucins, and the results presented strongly suggest that the informational content of mucin TRs is great and as of yet unexplored.

The mucin display platform is also ideal for the discovery and exploration of mucin-degrading enzymes such as the pathogenic glycoprotease StcE[46–49]. EHEC is a food-derived human pathogen able to colonize the colon and cause gastroenteritis and bloody diarrhea. Strains of the O157:H7 serotype carry a large virulence plasmid pO157:H7 that directs the secretion of StcE[46,51]. StcE is predicted to provide EHEC with adherence to the gastrointestinal tract and the ability to penetrate through the mucin layers via its impressive mucin-degrading properties[73]. StcE cleaves the C1 esterase inhibitor glycoprotein (C1-INH) that contains a highly O-glycosylated mucin-like domain and is required for complement activation[46]. StcE was previously shown to cleave several mucins including MUC1, MUC7, and MUC16[47,48,50,74], and the cleavage required O-glycosylation and accommodated complex O-glycan structures[48,75]. The gut microbiome is contained in a network of the gel-forming mucin MUC2 that forms the loose outer mucin layer, and a dense inner layer of MUC2 forms a barrier and prevents the microbiota from reaching the underlying colonic epithelium[76,77]. We found that

StcE efficiently binds to and cleaves MUC2 TRs, which would enable StcE to destroy the MUC2 networks and provide access to the epithelium (Fig. 5). However, we also discovered that the normal core3 O-glycosylation of MUC2 in the human colon[78–80], as well as the truncated cancer-associated STn glycosylation efficiently block cleavage by StcE. Currently, the molecular basis for the substrate selectivity of StcE with respect to glycoforms is unclear, but in this respect, it is perhaps relevant that the catalytic unit of StcE appears to exhibit high binding specificity for the dST core1 O-glycan[49]. Core3 O-glycosylation is restricted to the gastrointestinal tract in humans, and in the mouse, MUC2 is mainly glycosylated with core1 and core2 O-glycans that are also commonly found outside the gastrointestinal tract in humans[78,81,82]. Moreover, we found that the human MUC2 TR sequence was not amenable for core2 O-glycosylation in cells that introduced core2 O-glycans on other mucin TRs. Importantly, the core1 O-glycans installed on MUC2 TRs instead did not acquire the dST O-glycan with α2-6 sialylation as other mucin TRs (Fig. 4b), and we found that also this modification (dST/STn) blocked StcE cleavage (Fig. 5b,d). The normal core3 O-glycosylation pathway is downregulated in colorectal cancer[83], and truncated O-glycans including STn are found in colitis and other bowel inflammatory diseases[84]. Furthermore, mice deficient in the core3 b3gnt6 gene exhibit enhanced susceptibility to colorectal cancer and colitis[82,85]. These findings are highly relevant for the understanding of the functions of MUC2 in the intestine as a barrier and in the containment of the gut microbiome[78]. Coincidently, the α2-6 sialyltransferase ST6GALNAC1 is quite selectively and highly expressed in the intestine, and the expression pattern mimics that of the CASD1 enzyme that O-acetylates sialic acids and blocks the action of most sialidases[86]. Expression of 9-O-acetylated STn is remarkably specific for normal intestine[63], and while STn is generally considered a cancer-associated type of O-glycosylation, in fact, deacetylation of normal intestine demonstrates wide reactivity for STn in normal goblet cells producing MUC2[63]. We, therefore, propose that the O-glycosylation of MUC2 in goblet cells has co-evolved with MUC2 to provide protection from StcE-like mucin-degrading glycopeptidases by introducing the core3 pathway with ST6GALNAC1 mediated α2-6 sialylation and 9-O-acetylation by CASD1, and by the design of the TR sequence that inhibits core2 branching. We further predict that the unusual accumulation of 9-O-acetylated STn in goblet cells may be a result of overexpression of ST6GALNAC1, which could selectively block core1 synthesis and favor core3 synthesis.

The mucin display platform further enabled us to dissect the mucin-binding properties of StcE and discover a distinct mucin-binding module X409 on StcE (Fig. 6). Recently, the mucin-binding properties of StcE were explored with a catalytic inactivated mutant enzyme following the concept that inactivated hydrolases often can be used as binding reagent[52]. However, the results presented here demonstrate that the mucin-binding properties of StcE are conferred exclusively by the distinct X409 binding domain placed in the C-terminal region (Fig. 6). Importantly, the mucin-binding properties of X409 are not driven by a particular O-glycan structure suggesting a more complex interaction with the mucin TR backbone and innermost monosaccharide residues of attached O-glycans (Fig. 6e). Carbohydrate-binding modules (CBMs) are found widely on microbial glycoside hydrolases, glycosyltransferases, and other carbohydrate-active enzymes; in most cases, they serve in localizing these enzymes to their substrates, but sometimes participate in modulating the enzymatic functions[87,88]. Among eukaryotic glycosyltransferases, only the GALNTs directing O-glycosylation have appended CBMs, which orchestrate distant glycosylation by coordinating partially GalNAc-glycosylated substrates into the catalytic site[89]. The identified X409 mucin-binding module on StcE clearly directed the binding to O-glycosylated

mucin TRs with a preference for more complex O-glycosylation and interestingly slightly lower binding to the Tn and STn glycoforms. Further studies are needed to explore how X409 selectively binds mucin TRs with diverse O-glycan structures, and to determine if this binding domain may be classified as a CBM.

The sialic acid receptor specificity of influenza virus HA is essential for virus transmission[90], while the receptor destroying activity of NA that cleaves off sialic acid is required for release and propagation of the virus[91]. We previously demonstrated that the cell-based display of the human glycome could be used reliably to dissect the α2-3 and α2-6 sialic acid-binding specificities of influenza HAs and provide information on the underlying glycoconjugate nature[15]. Here, we extended these studies with PR8, a mouse-adapted IAV displaying α2-3Neu5Ac linkage specificity, and mucin TRs to explore the role of mucins in HA binding and NA release. IAV encounters mucins in secretions and on cell surfaces, and penetration of the mucus layer is a prerequisite for infection of respiratory tissues[55]. Interestingly, we found specific interactions of PR8 IAV with MUC1 and interactions of this α2-3Neu5Ac binding virus depended on the presence of core2 O-glycans. MUC1 was previously suggested to interact with IAV limiting binding to host epithelium and subsequent infection, and synthetic MUC1 peptides decorated with STn or ST reduced IAV infection of MDCK cells in vitro[92]. Our findings support the selective binding of PR8 to MUC1 with preferred binding specificities demonstrating the ability of the mucin display to address mucin and glycan context of influenza virus binding with an opportunity to produce decoy reporters for interference studies.

In summary, the cell-based mucin display platform provides a unique resource, that for the first time will enable deeper exploration of the nature and functions of human mucins. We illustrated this with classical examples of binding studies with microbial adhesins and substrate analysis of a microbial glycopeptidase that provided clear evidence of selectivities for different human mucins and their O-glycans. The finding that defined mucin TR modules can be produced with programmable O-glycans will enable the microbiome community to integrate mucins in studies at a level and detail not previously envisioned. In this respect, recent reports have demonstrated that sparse and heterogenous isolated human MUC5AC and O-glycans, therefore, have unique properties and trigger the downregulation of virulence genes and the disintegration of biofilms[93].

## Methods

**Cell culture**. HEK293-6E[WT] (obtained through a license agreement with Dr. Yves Durocher, Bioprocédés Institute de recherche en Biotechnologie, Montréal) and all isogenic clones were cultured in DMEM (Sigma-Aldrich) supplemented with 10% heat-inactivated fetal bovine serum (Sigma-Aldrich) and 2 mM GlutaMAX (Gibco) in a humidified incubator at 37 °C and 5% CO$_2$. HEK293-6E were also grown in suspension in serum-free F17 culture media (Invitrogen) supplemented with 0.1% Kolliphor P188 (SIGMA) and 4 mM GlutaMax at 37 °C and 5% CO$_2$ under constant agitation (120 rpm). All glycoengineered isogenic HEK293 cells used in this study are listed in Supplementary Table 2 and are available as part of the cell-based glycan array resource[15].

**CRISPR/Cas9-targeted KO in HEK293 cells**. CRISPR/Cas9 KO was performed using the GlycoCRISPR resource containing validated gRNAs libraries for targeting of all human glycosyltransferases[94]. In brief, HEK293 cells grown in 6-well plates (NUNC) to ~70% confluency were transfected for CRISPR/Cas9 KO with 1 µg of gRNA and 1 µg of GFP-tagged Cas9-PBKS using lipofectamine 3000 (Thermo-Fisher Scientific) following the manufacturer's protocol. Twenty-four hour post-transfection, cells were bulk-sorted based on GFP expression by FACS (SONY SH800). After one week of culture, the bulk-sorted cells were single cell-sorted into 96-well plates. KO clones were screened by Indel Detection by Amplicon Analysis (IDAA)[95] with the primers amplifying gRNA targeting sites and final clones were further verified by Sanger sequencing. All the gRNA and the primers used in this study were listed in Supplementary Table 3.

**Human mucin TR reporters**. The transmembrane mucin TR reporters were designed by fusion of human MUC1 signal peptide (amino acids 1–62, Uniprot P15921) with 6xHis, Flag-tag, EGFP, multiple cloning site, and the membrane anchoring domain of human MUC1 (amino acids 1042–1138) (Fig. 1 and Supplementary Table 1)[15,72]. Exchangeable mucin TR inserts of 150–200 amino acids were synthesized as TrueValue constructs with in-frame BamHI and NotI sites (Genewiz, USA). The secreted TR plasmids contain NotI/XhoI restriction sites and a 6xHis tag STOP encoding ds oligo (5′-GCGGCCGCCCATCACCACCATCAT CACTGATAGCGCTCGAG-3′, NotI/XhoI restriction sites underlined). We also included a TR reporter design containing six 11-mer sequences with a single O-glycosylation site (AEAAATPAPAK$_{n=6}$) to serve as a control for the patterns of O-glycans found in mucin TRs (Supplementary Fig. 1a and Supplementary Table 1).

**Transient transfection with mucin TR reporters**. Transmembrane GFP-tagged mucin TR reporters were transiently expressed in engineered HEK293 cells. Briefly, cells were seeded in 24-wells (NUNC) and transfected at ~70% confluency with 0.5 μg of plasmids using Lipofectamine 3000. Cells were harvested 24 h post-transfection and used for assays followed by flow cytometry analysis.

**Production and purification of recombinant mucin TR reporters**. The secreted reporters were stably expressed in isogenic HEK293-6E cell lines selected by two weeks of culture in the presence of 0.32 μg/mL G418 (Sigma-Aldrich) and two rounds of FACS enrichment for GFP expression. A stable pool of cells was seeded at a density of $0.25 \times 10^6$ cells/mL and cultured for 5 days on an orbital shaker in F17 medium (Gibco) supplemented with 0.1 Kolliphor P188 (Sigma-Aldrich) and 2% Glutamax. Culture medium containing secreted mucin TR reporter was harvested ($3000 \times g$, 10 min), mixed 3:1 (v/v) with 4× binding buffer (100 mM sodium phosphate, pH 7.4, 2 M NaCl), and run through a nickel-nitrilotriacetic acid (Ni-NTA) affinity resin column (Qiagen), pre-equilibrated with washing buffer (25 mM sodium phosphate, pH 7.4, 500 mM NaCl, 20 mM imidazole). The column was washed multiple times with washing buffer and mucin TR reporter was eluted with 200 mM imidazole. Eluted fractions were analyzed by SDS-PAGE and fractions containing the mucin TR reporter were desalted followed by buffer exchange to MiliQ using Zeba spin columns (ThermoFisher Scientific). Yields were quantified using a Pierce™ BCA Protein Assay Kit (ThermoFisher Scientific) following the manufacturer's instructions and NuPAGE Novex Bis-Tris (4–12%, ThermoFisher Scientific) Coomassie blue analysis.

**O-glycoprofiling**. HPLC (C4) purified mucin TR reporters (10 μg) were incubated in 0.1 M NaOH and 1 M NaBH$_4$ at 45 °C for 16 h. Released O-glycan alditols were desalted by cation-exchange chromatography (Dowex AG 50W 8X). Borate salts were converted into methyl borate esters by adding 1% acetic acid in methanol and evaporated under N$_2$ gas. Desalted O-glycan alditols were permethylated (in 150 μL DMSO, ~20 mg NaOH powder, 30 μL methyl iodide) at room temperature for 1 h. The reaction was terminated by addition of 200 μL ice-cold MQ water followed by the addition of ~200 μL chloroform. The organic phase was washed 5 times with 1 mL MQ water and evaporated under N$_2$ gas. Permethylated O-glycans were purified by custom Stage Tips (C18 sorbent from Empore 3 M) and eluted in 20 mL 35% (v/v) acetonitrile, of which 1 μL was co-crystalized with 1 μL DHB matrix (10 mg/mL in 70% acetonitrile, 0.1% TFA, 0.5 mM sodium acetate) before positive mode MALDI-TOF analysis.

**Isolation of mucin TR O-glycodomains**. Ni-chromatography purified intact mucin TR reporters (50 μg) were digested with 1 μg Lys-C (Roche) at a 1:35 ratio at 37 °C for 18 h in 50 mM ammonium bicarbonate buffer (pH 8.0). After heat inactivation at 98 °C for 15 min, reactions were dried by speed vac and desialylated with 40 mU *C. perfringens* neuraminidase (Sigma-Aldrich) for 5 h at 37 °C in 65 mM sodium acetate buffer (pH 5.0). This step was omitted for reporters expressed in HEK293$^{KO\ COSMC}$ (Tn glycoforms). Samples were heat-inactivated at 98 °C for 15 min and dried. For intact MS analysis samples were separated by C4 HPLC (Aeris™ C4, 3.6 μm, 200 a, 250 × 2.1 mm, Phenomenex) using a 0–100% gradient of 90% acetonitrile in 0.1% TFA. Fractions containing the released TR O-glycodomains were verified by ELISA with lectins or mAbs, dried, and resuspended in 20 μL of 0.1% FA for intact mass analysis. For bottom-up analysis of the MUC1 reporter, samples (20 μg) were further digested 2× with 0.67 μg Endo-AspN at a 1:35 ratio for 18 h at 37 °C in 100 mM Tris-HCL (pH 8.0). After inactivation by the addition of 1 mL of concentrated TFA, samples were desalted using custom Stage Tips (C18 sorbent from Empore 3 M) and analyzed by LC–MS/MS.

**O-glycopeptide bottom-up analysis of mucin TRs**. LC–MS/MS analysis was performed on EASY-nLC 1200 UHPLC (ThermoFisher Scientific) interfaced via nanoSpray Flex ion source to an Orbitrap Fusion Lumos MS (ThermoFisher Scientific). Briefly, the nLC was operated in a single analytical column set up using PicoFrit Emitters (New Objectives, 75 mm inner diameter) packed in-house with Reprosil-Pure-AQ C18 phase (Dr. Maisch, 1.9-mm particle size, 19–21 cm column length). Each sample was injected onto the column and eluted in gradients from 3 to 32% B for glycopeptides, and 10 to 40% for released and labeled glycans in 45 min at 200 nL/min (Solvent A, 100% H$_2$O; Solvent B, 80% acetonitrile; both containing 0.1% (v/v) formic acid). A precursor MS1 scan ($m/z$ 350–2000) of intact

peptides was acquired in the Orbitrap at the nominal resolution setting of 120,000, followed by Orbitrap HCD-MS2 and ETD-MS2 at the nominal resolution setting of 60,000 of the five most abundant multiply charged precursors in the MS1 spectrum; a minimum MS1 signal threshold of 50,000 was used for triggering data-dependent fragmentation events. Targeted MS/MS analysis was performed by setting up a targeted MSn (tMSn) Scan Properties pane.

**Intact mass analysis of mucin TRs**. Samples were analyzed by EASY-nLC 1200 UHPLC (ThermoScientific Scientific) interfaced via nanoSpray Flex ion source to an on OrbiTrap Fusion/Lumos instrument (ThermoScientific Scientific) using "high" mass range setting in $m/z$ range 700–4000. The instrument was operated in "Low Pressure" Mode to provide optimal detection of intact protein masses. MS parameters settings: spray voltage 2.2 kV, source fragmentation energy 35 V. All ions were detected in OrbiTrap at the resolution of 7500 (at $m/z$ 200). The number of microscans was set to 20. The nLC was operated in a single analytical column set up using PicoFrit Emitters (New Objectives, 75 mm inner diameter) packed in-house with C4 phase (Dr. Maisch, 3.0-mm particle size, 16–20 cm column length). Each sample was injected onto the column and eluted in gradients from 5 to 30% B in 25 min, from 30 to 100% B in 20 min and 100% B for 15 min at 300 nL/min (Solvent A, 100% H$_2$O; Solvent B, 80% acetonitrile; both containing 0.1% (v/v) formic acid).

**Data analysis**. Glycopeptide compositional analysis was performed from $m/z$ features extracted from LC–MS data using in-house written SysBioWare software[96]. For $m/z$ feature recognition from full MS scans Minora Feature Detector Node of the Proteome discoverer 2.2 (ThermoFisher Scientific) was used. The list of precursor ions ($m/z$, charge, peak area) was imported as ASCII data into SysBioWare and compositional assignment within 3 ppm mass tolerance was performed. The main building blocks used for the compositional analysis were: NeuAc, Hex, HexNAc, dHex, and the theoretical mass increment of the most prominent peptide corresponding to each potential glycosites. Upon generation of the potential glycopeptide list, each glycosite was rank for the top 10 most abundant candidates and each candidate structure was confirmed by doing targeted MS/MS analysis followed by manual interpretation of the corresponding MS/MS spectrum. For intact mass analysis raw spectra were deconvoluted to zero-charge by BioPharma Finder Software (ThermoFisher Scientific, San Jose) using default settings. Glycoproteoforms were annotated by in-house written SysBioWare software[96] using average masses of Hexose, N-acetylhexosamine, and the known backbone mass of mucin TR reporter sequence.

**Cell-binding assays**. For lectin staining HEK293 cells transiently expressing mucin TR reporters were incubated on ice or at 4 °C with biotinylated PNA (0.2 μg/mL), VVA (0.2 μg/mL) (Vector Laboratories) or Pan-lectenz (1.0 μg/mL) (Lectenz Bio) diluted in PBA (1× PBA containing 1% BSA (w/v)) for 1 h, followed by washing and staining with Alexa Fluor 647-conjugated streptavidin (1:1000) (Invitrogen by ThermoFisher Scientific) for 20 min. Stainings with mAbs specific to mucin glycoforms produced in mice were performed by incubating cells for 30 min at 4 °C with supernatant harvested from the respective hybridoma followed by staining with Alexa Fluor 647-conjugated goat anti-mouse IgG (1:1000) (Invitrogen by ThermoFisher Scientific) for 1 h. Cells were stained with GST-tagged strepto-coccal adhesins at 10 nM concentration diluted in PBA for 1 h on ice, followed by incubation with rabbit polyclonal anti-GST antibodies (1:500) (ThermoFisher) for 1 h and subsequent staining with Alexa Fluor 647-conjugated goat anti-rabbit IgG (1:300) (Invitrogen by ThermoFisher Scientific) for 1 h. All cells were resuspended in PBA for flow cytometry analysis (SONY SA3800). Mean fluorescent intensity (MFI) of the binding of streptococcal adhesins to GFP positive (expressing mucin TR reporters) and negative (not expressing) populations was quantified using FlowJo software (FlowJo LLC).

**ELISA**. ELISA assays were performed using MaxiSorp 96-well plates (Nunc) coated with dilutions of purified mucin TR reporters from 100 ng/mL or fractions derived from C4 HPLC incubated o/n at 4 °C in 50 mL carbonate-bicarbonate buffer (pH 9.6). Plates were blocked with PLI-P buffer (PO$_4$, Na/K, 1% Triton-X 100, 1% BSA, pH 7.4) and incubated with mAbs 3C9, 5F4 and TKH2 (undiluted culture supernatants), biotinylated-lectins VVA (0.5 μg/mL), PNA (0.5 μg/mL), MAL II (2.0 μg/mL) (Vector Laboratories) or Pan Lectenz (2.0 μg/mL) (Lectenz Bio) for 1 h at RT, followed by extensive washing with PBS containing 0.05% Tween-20, and incubation with 50 mL of 1 μg/mL HRP conjugated anti-mouse Ig (Dako) or 1 μg/mL streptavidin-conjugated HRP (Dako) for 1 h. Plates were developed with TMB substrate (Dako) and reactions were stopped by the addition of 0.5 M H$_2$SO$_4$ followed by measurement of absorbance at 450 nm.

**StcE proteolytic activity and binding assays**. *E. coli* codon optimization, gene synthesis and cloning of the recombinant StcE (residues 39-898), StcE$^{E447D}$ (residues 39-898 with mutation of E447D), StcE$^{ΔX409}$ (residues 39-796), and X409 (residues 797-898) were outsourced to Twist Bioscience (USA). The genes were cloned in a pet28-based vector (Kanamycin) on the direct 3′ end of a MHHHHHHSSHENLYFQG linker. The plasmids were transformed in T7 Express (NEB) bacterial strains, grown at 37 °C for 2 h, induced with 1 mM IPTG, and

cultures were continued at 16 °C overnight. Cells were harvested by centrifugation and lysed in buffer A (50 mM Tris-HCl, 300 mM NaCl, 10 mM imidazole, pH 8) supplemented with 0.5 mg/mL Lysozyme. Proteins were purified using Ni Sepharose 6 Fast Flow resin (GE Healthcare, Uppsala, Sweden) and eluted with buffer B (Buffer A + 250 mM imidazole). The purification was finished on a Superdex 200 16/60 gel-filtration column (GE Healthcare, Uppsala, Sweden) equilibrated in buffer A. Fractions containing enzyme were pooled, dialysed in PBS, and frozen. Enzyme assays with purified intact mucin reporters (500 ng) were performed by incubating serial dilutions of StcE for 2 h at 37 °C in 20 µL reactions in 50 mM ammonium bicarbonate buffer, and reactions were stopped by heat inactivation at 95 °C for 5 min. Samples were run on NuPAGE Novex gels (Bis-Tris 4–12%) at 100 V for 1 h followed by staining with Krypton Fluorescent Protein Stain (ThermoFisher Scientific) according to the manufacturer's instructions. Gels were imaged using an ImageQuant LAS 4000 system (GE Healthcare). Cell-based activity assays were performed with HEK293 cells transiently expressing mucin TR reporters incubated with serial dilutions of StcE in PBA at 37 °C. After 1 h, cells were washed with PBA, stained with APC-conjugated anti-FLAG antibody (1:1000) (BioLegend) for 30 min at 4 °C, and washed cells were analyzed by flow cytometry. MFI of anti-FLAG binding to GFP positive (transfected) and negative (untransfected) populations was quantified as using FlowJo. For cell-binding assays with X409-GFP, HEK293 cells expressing ECFP (Enhanced Cyan Fluorescent Protein) tagged membrane TR reporters were used. Cells were incubated with different concentrations of X409-GFP for 1 h at 4 °C followed by staining with APC-conjugated anti-FLAG antibody. X409-GFP binding to anti-FLAG positive cells was quantified using FlowJo software. For histology analysis, deparaffinized tissue microarray sections[97] were microwave treated for 20 min in sodium citrate buffer (10 mM, pH 6.0) for antigen retrieval followed by 1 h blocking with 1× PBS containing 5% BSA (w/v). Sections were incubated o/n at 4 °C with 5 µg/mL 6xHis-tagged StcE$^{WT}$, StcE$^{ΔX409}$, or StcE$^{E447D}$ followed by washing and subsequent 1 h incubation with first mouse anti-6xHis antibody (1:1000) (R&D systems) and second AF488-conjugated goat anti-mouse IgG (1:500) (Invitrogen by Thermo-Fisher). Sections stained with 2 µg/mL GFP-X409 were optionally sialidase treated and co-stained with mouse anti-MUC2 (PMH1) (undiluted culture supernatant) and donkey-anti-mouse IgG Cy3 (1:500) (Jackson ImmunoResearch). All samples were mounted with ProLong Gold Antifade Mountant with DAPI (Molecular Probes) and imaged using a Zeiss microscopy system followed by analysis with ImageJ (NIH).

**Virus–mucin interaction assay**. Biolayer interferometry analysis of virus–mucin interactions was performed using a BLI Octet Red 384 system (ForteBio). For the direct mucin-binding and self-elution assays, mucins were loaded to saturation to Ni-NTA sensors. In the binding assay, the association of PR8 virus (100 pM) was performed in the presence of OsC (10 µM). In the self-elution assay, virus loading levels were adjusted as such that similar binding levels were obtained. After the binding phase, OsC was removed by three short washes, and the release of virions from the sensors was monitored. In the binding competition assay, virus binding to streptavidin sensors loaded to saturation with biotinylated recombinant soluble LAMP1 produced in 293T cells[57] was monitored in the presence of OsC and absence or presence of different concentrations of the different mucins. In the mucin-driven elution assay, the ability of mucins to elute viruses from LAMP1-coated sensors in the presence of OsC was monitored and the kinetics of virion release was determined from the steepest part of the curve.

**Reporting summary**. Further information on research design is available in the Nature Research Reporting Summary linked to this article.

## Data availability
All data generated or analyzed during this study are included in this article and Supplementary Information Files. The mass spectrometry proteomics data have been deposited to the ProteomeXchange Consortium via the PRIDE repository with the dataset identifier PXD024851. Source data are provided with this paper.

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

## Acknowledgements

This work was supported by the Lundbeck Foundation, the Novo Nordisk Foundation, the European Commission (GlycoImaging H2020-MSCA-ITN-721297, BioCapture H2020-MSCA-ITN-722171), the Danish National Research Foundation (DNRF107), the Mizutani Foundation (to Y.N.), the European Union's Horizon 2020 Research and Innovation Program under the Marie Sklodowska-Curie grant agreement No 787684 (to C.B.), and the National Institutes of Health grant (R01GM32373 to A.V. and GM137458 to T.M.I.).

## Author contributions

R.N., C.B., H.C., and Y.N conceived and designed the study; A.K., L.S., Z.Y., A.H., D.M.S., S.F., U.M., L.H., S.Y.V., H.J.J., and L.A.D contributed with experimental data and interpretation; T.M.I., B.A.B., P.M.S., and A.V. contributed to the streptococcal adhesin studies; L.D., R.V., F.D., and B.H. contributed to the glycomucinase studies; W.D., E.V., and C.A.M.d.H. contributed to the influenza studies; R.N., C.B., H.C., and Y.N. wrote the manuscript, and all authors edited and approved the final version.

## Competing interests

The University of Copenhagen has filed a patent application relating to X409 mucin-binding peptides (EPO application EP21177857.6, pending). R.N., C.B., Y.N., B.H., and H.C. are named inventors. GlycoDisplay Aps, Copenhagen, Denmark, has obtained a license to the patent application. Y.N. and H.C. are co-founders of GlycoDisplay Aps and hold ownerships and financial interest in the company. The remaining authors declare no competing interests.
