## [Peer Review File · Nature Communications]

REVIEWER COMMENTS

Reviewer #1 (Remarks to the Author):

Nasson et al and the Clausen group has now used their engineered glycosyltransferase expression cells to decorate the tandem repeat sequences of all mucins using two reporters to generate mucin domain recombinant glycoproteins. The obtained glycoproteins has been characterized by mass spectrometry, antibody, lectin and effect of a glycoprotease, bacterial adhesins and influenza viral proteins.

These domains have been impossible to analyze due to technical reasons and thus we are lacking understanding how external mucosal surfaces look like and what commensal and pathogenic bacteria will encounter. The present tools are thus very welcome. They will provide us with novel approaches helping us in this difficult, unexplored, and important research area. A glimpse of what will be learnt is the different glycosylation of MUC2 TR and how this glycosylation likely will help to make it more resistant to a bacterial protease. This important work will be the basis a new development in the understanding of glycans as it can address the physiological binding to glycan surfaces (patches) in contrast to single glycan binding.

MAJOR

1. The major problem is the poor writing of especially the Result section. Poor language is frequently used, for example: the lab slang 'constructs' is repeated in almost every line, several times in two sentence in the last paragraph of page 4. The text should be written for a broader group of readers and not only those that know all the glycan 'nomenclature'. The T, Tn, mSTa, etc has to be explained at least when introduced in the text. Other sentences are impossible to understand (line 146). Some sections, like the influenza virus is very difficult to follow what has been done.

2. There is a common misunderstanding around PTS sequences that they are built by identical tandem repeats, something that is caused by the difficulties in sequencing long repeated DNA sequences. Most such domain are more or less degenerated and sometimes not even possible to reveal. This varies between the mucins and species as there is no sequence conservation. An example is human MUC2 that was sequenced for the first time 2018 (Svensson, Scientific Reports). The here used repeat sequences for PTS1 is only found in 4 out of 22 imperfect repeats and PTS2 in 26 of 98 repeats. The used repeats are the most common sequences and thus relevant and to call them TR. However, this should defined and explained in the introduction.

3. The title should be revised. 'Mucinome' is not defined and imprecise. One possibility is to replace it with 'mucin domain'.

4. Fig. 1. The membrane anchored reporter should also be shown. The figure to the left, KO and KI, does not contribute anything.

5. Fig. 2. Should be condensed into one page by removing empty areas. Overall for all figures, should the sizes of letters/numbers be controlled for size allowing a reader to read the figure after printing.

5. Fig. 3. The text, line 235 and in discussion, that MUC2 is 'almost exclusively decorated with Core 1' is an overstatement.

6. Fig. 4c. What is Ctrl and how could it be the same with and without StcE?

7. Text to Fig. 6. The equipment used is not stated. What is PR8? The rationale for using LAMP1 should be clarified.

Reviewer #2 (Remarks to the Author):

In this manuscript, Nason et al. create a library of mucins with defined O-glycans using a KO/KI approach in 293 cells used previously by Clausen and co-workers. The motivation comes from the fact that mucin O-glycosylation is extremely different to study given the sheer amount of glycans and, therefore, having mucins with defined glycoforms would enable the function of glycosylation to be investigated. Such mucins were successfully generated and partially characterized by the total amount of sialic acid-removed glycans by mass spectrometry. These membrane-bound and soluble mucins were used in three applications: (i) investigating binding to bacterial Siglec-like adhesins; (ii) examining the mode of action for the O-glycan dependent protease StcE; (iii) an influenza binding assay to probe HA/NA activities. Overall, this work is very interesting and provides tools needed to help further evaluate the functions of mucins. I have several questions/concerns/comments that may help improve the manuscript:

1. Is the X409-GFP construct monomeric? Is its affinity known for a glycan? Saturating binding at 10 ng/mL seems remarkable. Antibody binding to cells is not saturated until close to 1 ug/mL, which is

100-times higher despite likely have significantly higher affinity when accounting for their dimeric interaction. Therefore, unless X409 interactions are either multivalent or very high affinity, I'm having a hard time wrapping my head around saturating binding at such low concentrations.

2. Is the binding pattern of X409 and the catalytically inactive StcE the same? While the authors very nicely show that X409 contributes to binding as a carbohydrate-binding module, I don't think it is ruled out that the catalytic domain may influence the specificity/affinity.

3. Inhibition of StcE by Core3 and sialyl-Tn (Figure 4d) is interesting, but no model is presented on how exactly this is occurring. Are these outcompeting recognition of mucins by the X409 domain? If that was the case, I would have expected Core3 and sialyl-Tn to bind the strongest to X409 but based on Figure 5e that does not appear to be the case. Binding affinity measurements of X409 and a catalytically inactive StcE to the defined mucins would be an excellent addition to this section to help sort this out.

3. For the flow cytometry with lectin, the methods describe that biotinylated lectins were detected with streptavidin-AF488, while AF488 secondary was used for antibody detection. This seems like a very odd choice of fluorophore given the cells are expressing GFP that has very similar spectra properties as AF488. Could authors please double check this or clarify how this was done.

4. Similar to my concern above, in Fig 5d, is stated that binding of X409-GFP is to 293 cells expressing the TR reporters? Yet Fig 1 shows that the reporters have GFP. More details are needed here.

5. Could the authors describe why neuraminidase was used as this is remarkably missing. It would be helpful if the authors could include at least one spectra on what a mucin looks like prior to neuraminidase.

6. In the text, in addition to describing the MS carried out as 'in tact MS', could other describe specifically what type of MS it is.

7. Is total site occupancy changes significantly when different glycoforms are used? The certain types of glycosylation inhibit addition of O-glycans to other sites (or even having the opposite effect of causing a site to be modified).

8. There was no mention of the N-glycan on Muc22. Was this glycosylation expected and seen in the past?

9. In Fig 1, it shows that certain mucins are expressed as soluble proteins while others as membrane bound proteins. However, in Fig 2A the first three mucins are in the soluble class within Fig 1. This then happens throughout. I suggest some clarifying points in the text regarding Fig 1 to say that these were used as solution and membrane-bound. I bring this up because I found it very confusing on first pass.

Reviewer #3 (Remarks to the Author):

This manuscript comes from an established group that has been making various gene knockout cell systems for over a decade, particularly with focus on O-glycosylation pathways. Previous studies have presented major advances for the field, particularly using the SimpleCell technology. Thus, this is a leading group with unique technical expertise in the area of study.

The current manuscript extends work initiated in a recent *Molecular Cell* paper, where the group developed 50+ isogenic HEK knockouts for studies of lectin and virus binding. In this paper, they now, express a panel of cell-surface and soluble mucin (MUC) proteins on a subset of these cell lines. They find that the S/T site occupancy on these MUCs is very high in many cases and thus propose that these reagents could be a unique resource to study mucin biology. The paper is well done in terms of characterizing these reagents. The usage examples of these new tools is somewhat less impressive as it repeats from previous work. As the mucin proteins developed contain small, but perhaps, significant heterogeneity in glycan composition, more careful analysis may also be needed to draw conclusions using this system—as glycan shielding, pattern formation and minor contaminations could impact finding—particularly for the lectin binding studies. Specific comments:

1. It should be explained why HEK cells is a good platform for studies of host-microbiome interactions? Would it not be more important to knockout or over-express genes in the gut epithelial cells for studies of the gut microbiome, and in airway epithelial cells for influenza. These cells may express mucins at different/stoichiometric concentrations that impact outcome. In this regard, the theme of the paper revolves around the relevance of mucins in host-microbiome interactions, but there are no studies with the gut microbes in this paper—just adhesins and glycoproteases in reconstituted systems. The previous *Molecular Cell* paper already presents these reconstituted system studies using some of the model systems used in this paper—validation of some of the propositions in the more complex milieu would help.

2. Regarding, the characterization of the mucin tools. Why does anti-Tn-MUC4 mAb (3B11 and 6E3) bind MUC2#2 but not MUC4 (Supplementary Fig. 2)? There is a large discussion of mAb CLH2 in Results but no data are provided and the point the authors are making is not clear. mAb 2D is mentioned in Results but, again, data were missing?

3. The change on molecular mass (shift) in Supplementary Fig. 3 should be discussed since some mucins seem to increase mass when expressed in C1GalT1 cells (like MUC5AC, MUC2TR#1), while others like MUC4 drop. The increase in mass is puzzling. The extent of molecular mass reduction seems small in other cases since glycans often constitute a large part of mucinous protein mass. Is this what would be expected based on theoretical estimation of O-glycan mass?

4. If the mucin proteins are to be used for functional studies, there should be some confidence that these are uniform products or at least there should be clear documentation of heterogeneity so that ensuing results may be interpreted in that light. Here, it is stated or implied that KO GCNT1 ST3Gal1/2 cells express T-antigen uniformly. Then, why do they bind PNA lectin in Supplementary Fig. 4 as PNA only binds desialylated Type-3 LacNAc? Similar increase in PNA binding is also observed in MUC1/7 expressed in KO GCNT1 cells, and VVA binding is also high in a number of systems besides the COSMC-KO. Could the authors also treat the COSMC-KO cells with a pan-sialidase to confirm that there is no sialyl-Tn antigen (VVA should not increase?). Figure 2 legends states that MUC1-HEK293WT data are presented in Figure 2, but I did not find these data. Overall, additional clarifications are needed to verify the proposition that these mucins are being produced with unique glycan epitopes by cell glycoengineering. Looking at the lectin binding data and mass spectrometry results (Fig. 2), however, it would appear that, while there is a clear difference in glycan distribution in the mucins produced in the different engineered cell lines, the final product may not be uniform (there may still be a mixture of glycan epitopes). That being the case, it would help to quantitatively document the degree of heterogeneity in the different products in Table format, so that the functional results can be interpreted accordingly.

5. Figure 3: In a previous paper (ref. 15), the authors tested many more panels of knockouts for two of the adhesins discussed in this manuscript. Is there a reason why only one edited cell types is considered here? Also, please confirm that the base HEK cells (and HEK-KO GCNT1/ST6GalNAc clones) themselves do not bind the two new adhesins in this paper or express MUC proteins endogenously. There are also no data for MUC over-expression on these cells (using anti-FLAG) to ensure that the expression of all mucin proteins occurs at the same level. GFP may not be a good surrogate reporter for cell surface protein expression, as it could also be fluorescent when inside cells. Figure legend states that data are also presented for HEK KO GCNT1—but this is not seen in this figure? Finally, there is a long discussion about saccharide patches and their potential importance based on selectin-literature. but there is less discussion about glycan shielding effects--- in this context the presence of a glycan itself appears to be insufficient for adhesin binding since other carbohydrates should also be absent at the same time for good binding—thus there seems to be competing effects that need to be accounted for.

6. A number of groups have previously studied the O-glycopeptidase StcE: enzymatic activity, substrate specificity, structural studies showing an inhibitory role for sialyl-Tn in its activity, and that this reagent can be used as a lectin following mutation. The current study expands on this using the

glycoengineered mucin reagents. The high/differential binding to pancreatic and breast tumors using X409 is novel and potentially important. These data seem to be part of Supplemental Data only.

7. The last figure shows that Influenza A can bind sialylated epitopes on mucins. It is not clear here if the binding depends on the simple expression of the glycan (which may be differentially expressed on various mucin TRs) or if there are other mucin/TR specific effects. Wouldn't similar binding not be observed if the biosensor was coated with synthetic glycans in the absence of the mucin? The reason for including these data in this manuscript is not entirely clear.

Point-by-point response & action list to reviewer comments

Reviewer #1

Query #1: The major problem is the poor writing of especially the Result section. Poor language is frequently used, for example: the lab slang ‘constructs’ is repeated in almost every line, several times in two sentences in the last paragraph of page 4.

The text should be written for a broader group of readers and not only those that know all the glycan ‘nomenclature’. The T, Tn, mSTa, etc has to be explained at least when introduced in the text.

Other sentences are impossible to understand (line 146).

Some sections, like the influenza virus is very difficult to follow what has been done.

Response #1: Thank you, we agree! It is important to maintain consistent reference to the mucin constructs as reporters. This is because they clearly do not reflect the entire mucin, and as pointed out by the reviewer (Query #2) TRs are imperfect in repeat sequences.

Action #1: We have tried to improve readability of the manuscript throughout and reduced the use of repeated wording like “constructs”. Constructs have been replaced in most cases by simply “mucin TR reporters” or “mucin TRs”,

The use of standard glycan nomenclature has been emphasized as follows in the Results section p.5 when first introduced:

“We took advantage of our previously reported O-glycoengineering strategy to establish designs for homogeneous O-glycosylation capacities that result in attachment of defined O-glycan structures (Fig. 1). The gene engineering included designs for O-glycans designated Tn (KO C1GALT1), STn (KO COSMC/KI ST6GALNAC1), T (KO GCNT1, ST3GAL1/2, ST6GALNAC2/3/4), monosialyl-T (mSTa) (KO GCNT1, ST6GALNACT2/3/4), as well as ST comprised of a mixture of mSTa and disialyl-T (dST) (KO GCNT1), with the O-glycan structures and genetic regulation illustrated in Figure 1.”

Query #2: There is a common misunderstanding around PTS sequences that they are built by identical tandem repeats, something that is caused by the difficulties in sequencing long repeated DNA sequences. Most such domain are more or less degenerated and sometimes not even possible to reveal. This varies between the mucins and species as there is no sequence conservation. An example is human MUC2 that was sequenced for the first time 2018 (Svensson, Scientific Reports). The here used repeat sequences for PTS1 is only found in 4 out of 22 imperfect repeats and PTS2 in 26 of 98 repeats. The used repeats are the most common sequences and thus relevant and to call them TR. However, this should defined and explained in the introduction.

Response #2: Thank you, we agree and this was why we included Suppl. Fig. 1 showing the complete sequence of the inserts used for the mucin reporters.

Action #2: The following sentences have been added to the Introduction, Results, and the legend for Supplementary Figure 1 to further clarify this:

p.2:

“A common characteristic of all mucins is that the major part of their extracellular region is comprised of variable number of imperfect tandem repeated (TR) sequences that carry dense O-glycans (Fig. 1).”

p.4:

“Figure 1 presents an overview of the concept for the cell-based display and production of human mucin TRs with programmed O-glycan structures, and Supplementary Figure 1 shows the entire sequences of the selected imperfect mucin TRs included in the TR reporter design. Supplementary Figure S1 illustrates how the TRs can produce characteristic patterns of O-glycans despite the imperfect sequence variations of TRs.”

p.29:

Supplementary Fig. 1a “Schematic presentation of the imperfect TR amino acid sequences selected for design of the human mucin TR reporters. All Ser/Thr residues are highlighted as potential O-glycosites by glycan symbols (mSTa O-glycan) to illustrate the characteristic patterning generated with all Ser/Thr residues O-glycosylated.”

Supplementary Fig. 1b “Parallel plot of key amino acid residues in the human mucin TR reporter designs. Residue combinations of TSP, TS and individual T, S, P and E counts in the different TR sequences represented in the mucin reporters are shown as residue count per sequence total.”

Query #3: The title should be revised. ‘Mucinome’ is not defined and imprecise. One possibility is to replace it with ‘mucin domain’.

Response #3: We disagree, but now define the word “Mucinome”.

Action #3: The following definition is included in the third paragraph of the Introduction:

“Thus, there are rich opportunities for unique codes in mucin TRs governed by the particular display of patterns and structures of O-glycans. The mucin TRs and their glycocodes may be considered the informational content of mucins and comprise the mucinome, which provides a much greater potential binding epitome than the comparatively limited repertoire of binding epitopes comprised of simple oligosaccharide motifs available in humans²².”

Query #4: Fig. 1. The membrane anchored reporter should also be shown. The figure to the left, KO and KI, does not contribute anything.

Response #4: This comment may be based on misunderstanding. The left side of Figure 1 is important in order to illustrate that the genetic engineered involves targeted KO/KI, and the center part illustrates both membrane bound and the secreted mucin TR reporters.

Action #4: The Figure 1 legend has been revised as follows to stress these points:

Fig. 1 Design of the human mucin tandem repeat (TR) display platform. Illustration of the mucin TR display approach with membrane bound and secreted mucin reporters expressed in KO/KI glycoengineered isogenic HEK293 cell lines. HEK293 wild type (WT) cells are predicted to produce a mixture of mSTa, dST and sialylated core2 structures, and through stable genetic engineering a library of isogenic HEK293 cells with different O-glycosylation capacities were developed. These cells enable display of mucin TRs with different O-glycan structures as indicated (glycan symbols and genetic design shown) as well as tunable site occupancy by engineering of the *GALNT* isoenzyme gene repertoire (left part). The membrane and secreted mucin reporter construct designs share N-terminal 6xHis and FLAG tags and GFP followed by different mucin TR domains of ca. 200 amino acids (single TR domains used for MUC3, MUC5B, MUC13, MUC6 and GP1b α). The cell membrane construct design further includes

the SEA and transmembrane domain of human MUC1 in the C-terminal for membrane retention, while the secreted construct design has a second C-terminal 6xHis tag. The most characteristic TR sequence for each construct is illustrated with the number of TRs included (right part). Full sequences of the TRs are shown in **Supplementary Figure 1**. Transient or stable expression of the mucin TR reporters in the glycoengineered isogenic HEK293 cell library enables display of cell surface mucin TRs as well as production of secreted mucin TRs with distinct O-glycan structures. Structures of glycans are shown with symbols drawn according to the Symbol Nomenclature for Glycans (SNFG) format⁹⁶.”

Query #5: Fig. 2. Should be condensed into one page by removing empty areas. Overall for all figures, should the sizes of letters/numbers be controlled for size allowing a reader to read the figure after printing.

Response #5: Thank you.

Action #5: All Figures have been condensed as much as possible and text size was increased.

Query #6: Fig. 3. The text, line 235 and in discussion, that MUC2 is ‘almost exclusively decorated with Core 1’ is an overstatement.

Response #6: Agree.

Action #6: The text p.8 has been revised to:

“The most remarkable finding was that the MUC2 TRs (especially the MUC2 region designated #1) had a relatively high proportion of core1 O-glycans with only α 2-3 linked sialic acids (mSTa) installed in HEK293^{WT} cells, while most other mucin TRs displayed a mixture of core2 and dST (Fig. 3b).”

Query #7: Fig. 4c. What is Ctrl and how could it be the same with and without StcE?

Response #7: Thank you. We inadvertently failed to describe the design of the control (Ctrl) reporter. We designed an artificial mucin TR sequence without dense O-glycosylation to demonstrate that StcE activity depends on densely O-glycosylated domains. This control reporter has the same GFP and FLAG tag as the mucin TR reporters and only a single Thr glycosite in the TR sequence. StcE does not cleave this control showing that StcE requires dense O-glycan patterns.

Action #7: The control TR sequence has been added to Supplementary Fig. 1a and Supplementary Table 2. The following text elements are now included:

In the Methods section:

“We also included a TR construct design containing six 11-mer sequences with a single O-glycosylation site (AEAAATTPAPAK_{n=6}) to serve as control for the patterns of O-glycans found in mucin TRs (**Supplementary Fig. 1a** and **Supplementary Table 1**)”

In the Results section:

“StcE efficiently cleaved most of the mucin TRs with the notable exception of MUC1 and MUC20, as well as the control TR construct designed with a single O-glycosite (**Fig. 4c** and **Supplementary Fig. 7c, d**).”

In Figure 4c legend:

“An artificially designed TR reporter with a single O-glycosite was used to serve as control (ctrl) for the clusters and patterns of O-glycans found in human mucin TRs.”

Query #8: Text to Fig. 7 The equipment used is not stated. What is PR8? The rationale for using LAMP1 should be clarified.

Response #8: Thank you.

Action #8: The text has been revised to read as follows to clarify:

Results p.10:

“Here, we addressed if the sequence and glycosylation of the mucin TRs influenced IAV HA-NA activity using biolayer interferometry^{55,56} by analyzing binding and dissociation kinetics of the laboratory mouse-adapted influenza strain A/Puerto Rico/8/1934 (H1N1) (PR8) virions (α 2-3Neu5Ac linkage specificity) to mucin-loaded sensors.”

and

“Next, the different mucin TR reporters were assessed for their ability to compete with PR8 virion binding to sensors coated with recombinant soluble lysosomal-associated membrane glycoprotein 1 (LAMP1), which carries multiple N-glycans with α 2-3 and α 2-6 linked sialic acids and serves as a potent IAV receptor⁶⁰.”

Materials and Methods p.21

“Biolayer interferometry analysis of virus-mucin interactions was performed using a BLI Octet Red 384 system (ForteBio), essentially similarly as described previously⁵⁶.”

Reviewer #2

Query #1: Is the X409-GFP construct monomeric? Is its affinity known for a glycan? Saturating binding at 10 ng/mL seems remarkable. Antibody binding to cells is not saturated until close to 1 μ g/mL, which is 100-times higher despite likely have significantly higher affinity when accounting for their dimeric interaction. Therefore, unless X409 interactions are either multivalent or very high affinity, I'm having a hard time wrapping my head around saturating binding at such low concentrations.

Response #1: The X409-GFP construct appears to be monomeric as determined by intact MS (not shown). There is no evidence that X409 binds a glycan and it does not bind to HEK293 cells without expressing distinct mucin TR reporters, and the binding studies here to mucin TRs with many different types of O-glycans suggest that the interaction may include both the peptide and glycan moieties. However, we thank the reviewer for pointing us to an inadvertent error in the labelling of concentrations of X409-GFP used for binding in Figure 5d – this should read [μ g/ml], and not ng/ml.

Action #1: Figure 5d has been revised accordingly.

Query #2: Is the binding pattern of X409 and the catalytically inactive StcE the same? While the authors very nicely show that X409 contributes to binding as a carbohydrate-binding module, I don't think it is ruled out that the catalytic domain may influence the specificity/affinity.

Response #2: We disagree. StcE without the X409 domain does not show binding to the tissues or the mucin TRs tested, while still being active in the cleavage assays (Fig. 5b and Supplementary Fig. 8a). However, as discussed, the full catalytically inactive StcE (E447D mutant) binds dST on printed glycan arrays (Yu *et al.* Structure 2012, cited ref 48) as well as mucins (Shon *et al.* PNAS, cited ref 51). Since the mucin binding properties of X409 seems rather independent of the structures of O-glycans on the mucin TRs (Fig. 5e), it is likely that the previous observed binding to dST O-glycans on printed glycan arrays is driven by the catalytic unit. This interpretation may also explain recent preliminary and controversial findings that affinity chromatography with the StcE-E447D mutant containing both the catalytic unit and X409 enriched not only human mucins but also a wide range of O-glycoproteins without clear mucin domains (Malaker *et al.* 2021, doi: <https://doi.org/10.1101/2021.01.27.428510>).

Action #2: We have now included the inactive mutant StcE^{E447D}. The following text has been modified to clarify this:

p. 8 Results:

“**The glycoprotease StcE** - Secreted protease of C1 esterase inhibitor (StcE) from Enterohemorrhagic *Escherichia coli* (EHEC) O157:H7 is a zinc metalloprotease with remarkable ability for cleaving densely O-glycosylated mucins and mucin-like glycoproteins. StcE is thought to serve in colonic mucin degradation facilitating EHEC adherence to the epithelium and subsequent infection^{45,46}. Recent studies demonstrated that StcE has selective substrate specificity for S/T-X-S/T motifs with a requirement for O-glycans at the first S/T residue⁴⁷. Moreover, a catalytically inactive mutant of StcE (StcE^{E447D}) has glycan-binding properties for the dST core1 O-glycan as evaluated by printed glycan arrays⁴⁸, but also exhibited general binding properties for mucins⁴⁷⁻⁵⁰. We used the TRs displayed on cells to further explore the fine substrate specificity of recombinant purified StcE and to dissect its mucin-binding properties (**Fig. 4a**).”

p. 9 Results:

“Next, we explored the intriguing suggestion that StcE plays a role in adherence of EHEC to the intestinal epithelium by binding mucins^{46,48,52}. Examination of the 3-D structure of StcE⁴⁸ revealed that the protein has a C-terminal domain (here designated X409) opposite to the catalytic metalloprotease domain (M66), which we predicted could represent an evolutionarily mobile binding module. In line with previous findings, deletion of this X409 module did not affect the catalytic activity of StcE with the WT MUC2 reporter **Supplementary Fig. 8a**)⁴⁸, and the X409 construct alone did not exhibit enzymatic activity (**Fig. 5b**). However, deletion of the X409 module completely abrogated StcE binding to mucin producing cells in human colonic tissues, whereas the X409 alone (GFP-tagged) reproduced the strong tissue staining properties of the StcE^{WT} (**Fig. 5c**). Note also that the binding of active StcE, inactive StcE^{E447D}, as well as X409 alone fully recapitulate the tissue binding found previously with the inactive StcE^{E447D} mutant^{47, 49-51}.

Interestingly, the StcE/X409 staining appeared to overlap with MUC2 expression in human colon as shown by co-localization of staining with a mAb directed against GalNAc-glycosylated MUC2 (PMH1) (**Supplementary Fig. S8b**)²⁹. The X409 module displayed strong binding to normal human colon and stomach tissues and colon and stomach cancers. While we found low binding to other normal tissues including pancreas and breast, strong binding to the counterpart cancer tissues was observed (**Supplementary Fig. S8c**). Probing the X409 module with the cell-based mucin TR display revealed that this binding module bound to select human mucin TRs (carrying core2 O-glycans), but interestingly not to MUC1 (**Fig. 5d**). Moreover, the binding to MUC2, MUC5AC and to a lesser extent MUC7, was only

slightly influenced by the O-glycan structures attached to the TRs (**Fig. 5e**), with apparent weaker binding to TRs carrying Tn, core3 and especially STn O-glycans. These results clearly demonstrate that the X409 module mediates the mucin-binding properties of StcE, while the catalytic unit of StcE may mediate selective binding to dST O-glycans⁴⁸. Given that StcE, StcE^{E447D} and X409 alone exhibited the same highly selective tissue binding to mucin producing cells, it is likely that the dST glycan binding properties of the catalytic unit of StcE found with high density of glycans printed on glass slides⁴⁸, does not contribute substantially to binding to tissues. In conclusion, our results show that X409 does not directly impact the catalytic activity of StcE, and it is likely that this domain serves to target StcE to mucins through its highly selective mucin-binding properties^{46,48}. The X409 module may be exploited to probe for expression of mucins in normal and cancer tissues.”

p.14 Discussion

“Recently, the mucin-binding properties of StcE were explored with a catalytic inactivated mutant enzyme following the concept that inactivated hydrolases often can be used as binding reagent^{51,53}, however, the results presented here demonstrate that the mucin-binding properties of StcE are conferred exclusively by the distinct X409 binding domain placed in the C-terminal region (**Fig. 5**). Importantly, the mucin-binding properties of X409 are not driven by a particular O-glycan structure suggesting a more complex interaction with the mucin TR backbone and inner most monosaccharide residues of attached O-glycans (**Fig. 5e**).”

Query #3: Inhibition of StcE by Core3 and sialyl-Tn (Figure 4d) is interesting, but no model is presented on how exactly this is occurring. Are these outcompeting recognition of mucins by the X409 domain? If that was the case, I would have expected Core3 and sialyl-Tn to bind the strongest to X409 but based on Figure 5e that does not appear to be the case. Binding affinity measurements of X409 and a catalytically inactive StcE to the defined mucins would be an excellent addition to this section to help sort this out.

Response #3: This comment may be based on a misunderstanding. We clearly show that the X409 mucin-binding domain does not affect the cleavage rate and/or specificity of StcE in assays with mucin TRs in solution and/or displayed on cells (Fig. 5a,b and Supplementary Fig. 8).

Action #3: The following text p.13 in Discussion has been modified to clarify:

“However, we also discovered that the normal core3 O-glycosylation of MUC2 in the human colon⁷⁴⁻⁷⁶ as well as the truncated cancer-associated STn glycosylation efficiently block cleavage by StcE. Currently, the molecular basis for the substrate selectivity of StcE with respect to glycoforms is unclear, but in this respect it is perhaps relevant that the catalytic unit of StcE appears to exhibit high binding specificity for the dST core1 O-glycan⁴⁸.”

Query #4: For the flow cytometry with lectin, the methods describe that biotinylated lectins were detected with streptavidin-AF488, while AF488 secondary was used for antibody detection. This seems like a very odd choice of fluorophore given the cells are expressing GFP that has very similar spectra properties as AF488. Could authors please double check this or clarify how this was done.

Response #4: Thank you, typo error.

Action #4: Methods p.20 now reads:

“For lectin staining HEK293 cells transiently expressing mucin TR recombinant proteins were incubated on ice or at 4°C with biotinylated PNA, VVA (Vector Laboratories) or Pan-lectenz (Lectenz Bio) diluted in PBA (1x PBA containing 1% BSA (w/v)) for 1 h, followed by washing and staining with Alexa Fluor 647-conjugated streptavidin (Invitrogen) for 20 min.”

Query #5: Similar to my concern above, in Fig 5d, is stated that binding of X409-GFP is to 293 cells expressing the TR reporters? Yet Fig 1 shows that the reporters have GFP. More details are needed here.

Response #5: This is a misunderstanding as CFP-tagged TR reporters were used to detect binding of GFP-tagged X409.

Action #5: A cartoon illustration of the assay design has been added to Figure 5. The Methods section p21 has been reworded to clarify this:

“For cell-binding assays with X409-GFP, HEK293 cells expressing CFP-tagged membrane TR reporters were used. Cells were incubated with different concentrations of X409-GFP for 1 h at 4°C followed by staining with APC-conjugated anti-FLAG antibody. X409-GFP binding to anti-FLAG positive cells was quantified using FlowJo software.”

In Figure 5 legend the text was reworded to clarify:

“**d**, Binding of increasing concentrations of X409-GFP probe to HEK293^{WT} cells expressing mucin TR reporters tagged with CFP.”

Query #6: Could the authors describe why neuraminidase was used as this is remarkably missing. It would be helpful if the authors could include at least one spectra on what a mucin looks like prior to neuraminidase.

Response #6: Thank you. Neuraminidase treatment was used to reduce heterogeneity,

Action #6: The following text has been included p.6 in Results section:

“We used LysC digestion to liberate the intact TR O-glycodomains and C4 and C18-HPLC to purify these for further analysis (**Supplementary Fig. 5**). For direct intact mass analysis of mucin TRs we used pretreatment with neuraminidase to reduce complexity and facilitate deconvolution and interpretation.”

And

“For the bottom up analysis of the MUC1 TR reporter we also had to use pretreatment with neuraminidase because the sialylated glycoforms were poorly digested by AspN.”

Query #7: In the text, in addition to describing the MS carried out as ‘in tact MS’, could other describe specifically what type of MS it is.

Response #7: We agree, and this information was only described in the Methods section.

Action #7: We have now expanded the text and included the following in the Results section p.6:

“We used Lys-C to cleave the purified GFP-tagged MUC1 TR reporter and isolate the O-glycodomain without GFP, and applied LC-MS for intact mass analysis (**Fig. 2a**).”

Query #8: Is total site occupancy changes significantly when different glycoforms are used? The certain types of glycosylation inhibit addition of O-glycans to other sites (or even having the opposite effect of causing a site to be modified).

Response #8: We are uncertain what the reviewer specifically refers to, but the question being addressed is whether or not the O-glycan elongation process can interfere with the GalNAc initiation process. This is an important question as both initiation and elongation processes occur in Golgi, and the GALNTs that direct the initiation process utilize GalNAc-binding lectin domains for select substrate sites. If the first attached GalNAc residues are elongated the lectin-binding is blocked, and this may potentially reduce the number of O-glycans attached. We believe this was explained in detail on p.11.

Action #8: None

Query #9: There was no mention of the N-glycan on Muc22. Was this glycosylation expected and seen in the past?

Response #9: To our knowledge there are no experimental data reported on potential N-glycosites in mucin TRs, just as analysis of O-glycosites for most mucins are missing as discussed. We did verify that the N-glycosite in fact was N-glycosylated, but we do not believe that this information adds to the current manuscript. We intend to present the data in a follow-up study.

Action #9: None

Query #10: In Fig 1, it shows that certain mucins are expressed as soluble proteins while others as membrane bound proteins. However, in Fig 2A the first three mucins are in the soluble class within Fig 1. This then happens throughout. I suggest some clarifying points in the text regarding Fig 1 to say that these were used as solution and membrane-bound. I bring this up because I found it very confusing on first pass.

Response #10: Agree, thank you.

Action #10: The labeling in Figure 1 has been clarified, so it is clear that the Figure illustrates that we use membrane and secreted mucin TR reporters (central part) derived from the natural secreted and membrane bound mucins.

Reviewer #3

Query #1: It should be explained why HEK cells is a good platform for studies of host-microbiome interactions? Would it not be more important to knockout or over-express genes in the gut epithelial cells for studies of the gut microbiome, and in airway epithelial cells for influenza. These cells may express mucins at different/stoichiometric concentrations that impact outcome. In this regard, the theme of the paper revolves around the relevance of mucins in host-microbiome interactions, but there are no studies with the gut microbes in this paper—just adhesins and glycoproteases in reconstituted systems. The previous *Molecular Cell* paper already presents these reconstituted system studies using some of the model systems used in this paper—validation of some of the propositions in the more complex milieu would help.

Response #1: This comment does not appear to take state-of-the-art in the field into account. Moreover, our previous study in *Molecular Cell* does not address mucins.

The so-called reconstituted system or what we prefer to designate cell-based display of the human mucinome, for the first time enables production and display of the human mucin TR regions with different O-glycan structures and the full informational content. This is a major advance that cannot easily be replicated in any cell line, and there are really no advantages of doing so at this stage as the primary goal is to make mucin TRs available and to dissect their informational content in controlled experiments. We do provide illustrative examples for how these mucin TRs can be used, and we clearly - as the Reviewer does - dream of taking the new resource for defined mucin TRs into more complex biological systems. We believe that the manuscript clearly states the advances achieved and the concept behind making defined mucin TRs available to the Community.

Action #1: None

Query #2: Regarding, the characterization of the mucin tools. Why does anti-Tn-MUC4 mAb (3B11 and 6E3) bind MUC2#2 but not MUC4 (Supplementary Fig. 2)? There is a large discussion of mAb CLH2 in Results but no data are provided and the point the authors are making is not clear. mAb 2D is mentioned in Results but, again, data were missing?

Response #2: Thank you very much, typo error.

Action #2: A revised Supplementary Figure 2 is included. We chose to exclude the discussion of mAb CLH2 as this does not add substantial information. We have now corrected the typo in the Results section p.6:

“We also screened a larger panel of available anti-MUC1 mAbs for reactivity with different glycoforms of MUC1, which confirmed specific reactivities with defined glycoforms (Tn-MUC1 for 5E5 and 2D9; T-MUC1 for 1B9), and selective reactivity with Tn O-glycoforms (SM3 and HMGF1)(Supplementary Fig. 2c)”.

Query #3: The change on molecular mass (shift) in Supplementary Fig. 3 should be discussed since some mucins seem to increase mass when expressed in C1GalT1 cells (like MUC5AC, MUC2TR#1), while others like MUC4 drop. The increase in mass is puzzling. The extent of molecular mass reduction seems small in other cases since glycans often constitute a large part of mucinous protein mass. Is this what would be expected based on theoretical estimation of O-glycan mass?

Response #3: The migration of glycoproteins and in particular O-glycoproteins with high density of glycans such as mucins is essentially unpredictable and not directly related to mass, but more to the content of sialic acids. We did not intend to dissect this in detail and included Supplementary Figure 3 to illustrate that rather homogenous pure mucin TR reporters were produced.

Action #3: To clarify this we have reworded the following text p.6 in Results:

“Secreted mucin TR reporters stably expressed in glycoengineered HEK293 cells were isolated by Ni-chromatography and assessed by SDS-PAGE analysis, which showed that the GFP-tagged proteins migrated as distinct rather homogeneous bands (**Supplementary Figs. S3 and S4**).”

Query #4: If the mucin proteins are to be used for functional studies, there should be some confidence that these are uniform products or at least there should be clear documentation of heterogeneity so that ensuing results may be interpreted in that light. Here, it is stated or implied that KO GCNT1 ST3Gal1/2 cells express

T-antigen uniformly. Then, why do they bind PNA lectin in Supplementary Fig. 4 as PNA only binds desialylated Type-3 LacNAc? Similar increase in PNA binding is also observed in MUC1/7 expressed in KO GCNT1 cells, and VVA binding is also high in a number of systems besides the COSMC-KO. Could the authors also treat the COSMC-KO cells with a pan-sialidase to confirm that there is no sialyl-Tn antigen (VVA should not increase?). Figure 2 legends states that MUC1-HEK293WT data are presented in Figure 2, but I did not find these data. Overall, additional clarifications are needed to verify the proposition that these mucins are being produced with unique glycan epitopes by cell glycoengineering. Looking at the lectin binding data and mass spectrometry results (Fig. 2), however, it would appear that, while there is a clear difference in glycan distribution in the mucins produced in the different engineered cell lines, the final product may not be uniform (there may still be a mixture of glycan epitopes). That being the case, it would help to quantitatively document the degree of heterogeneity in the different products in Table format, so that the functional results can be interpreted accordingly.

Response #4: Thank you!! Supplementary Fig. 4 was prepared from antigen titrations and we chose to present a snapshot of the complete ELISA results to simplify. However inadvertently the single data point presentation was completely corrupted due to erratic shifting after pdf conversion of file. The statement in the legend to Figure 2 of intact MS analysis of MUC1 produced in HEK293WT cells was an error as this was tried, but not successful. We believe that the new representation of the full ELISA binding data with purified MUC1 TR reporters clarify the reviewers concerns regarding the glycosylation pathways engineered. The data presented now correlates with previous results¹⁵ and the flow cytometry results presented in Supplementary Figure 2. Please note that PNA is reported to bind Gal β 1-3GalNAc α -1-Ser/Thr accepting an inner α 2-6 linked sialic acid (mSTb).

Action #4: A new Supplementary Figure 4b is included with the entire ELISA data using antigen titration, and the data validates the predicted glycoforms.

Query #5: Figure 3: In a previous paper (ref. 15), the authors tested many more panels of knockouts for two of the adhesins discussed in this manuscript. Is there a reason why only one edited cell types is considered here? Also, please confirm that the base HEK cells (and HEK-KO GCNT1/ST6GalNAc clones) themselves do not bind the two new adhesins in this paper or express MUC proteins endogenously. There are also no data for MUC over-expression on these cells (using anti-FLAG) to ensure that the expression of all mucin proteins occurs at the same level. GFP may not be a good surrogate reporter for cell surface protein expression, as it could also be fluorescent when inside cells. Figure legend states that data are also presented for HEK KO GCNT1—but this is not seen in this figure? Finally, there is a long discussion about saccharide patches and their potential importance based on selectin-literature, but there is less discussion about glycan shielding effects--- in this context the presence of a glycan itself appears to be insufficient for adhesin binding since other carbohydrates should also be absent at the same time for good binding—thus there seems to be competing effects that need to be accounted for.

Response #5: - We previously assessed the preferred glycan structures of two of these adhesins using the GPIb α reporter protein as a backbone for the glycan display¹⁵. This led to identification of mSTa as the preferred glycan epitope for Hsa_{BR} and disialyl-core2 for 10712_{BR}. Here, we wanted to extend this by exploring how the mucin TR backbone influences binding by using the mucin TR display and we limited this study to the preferred mST and disialyl-core2 glycoforms. There are no reports on expression of endogenous mucins in HEK293 cells, and analysis of RNAseq data in public databases appear to confirm this. We also

included a new adhesin (*Streptococcus gordonii* GspB_{BR}), but decided to leave out the mutant of *Streptococcus mitis* (10712_{BR}).

Glycans may certainly shield/mask e.g. binding to protein epitopes and penultimate glycan structures. In the context of mucin TRs with different O-glycan structures and sites it is more likely that the binding context is provided by clustered presentation and patches which is supported by our findings that recognition of the adhesins is rather driven by the mucin TR sequence than removal of shielding/masking events. Importantly, the binding preference of these adhesins is to capping sialic acids which further speaks against potential unmasking/shielding effects.

Action #5: The following text in Results p.7 addresses the justification of the current study of adhesins:

“We previously used the cell-based glycan array platform to dissect the binding specificities of two serine-rich repeat (SRR) adhesins with Siglec-like binding regions expressed by oral streptococci, and demonstrated distinct binding specificities of *Streptococcus gordonii* (Hsa_{BR}) for mSTa and *Streptococcus mitis* (10712_{BR}) and disialylated core2 O-glycans displayed on a reporter containing the mucin-like domain in GP1b α (**Fig. 1**)^{15,40,41}. We also found evidence that these adhesins showed binding selectivity for mucin TRs expressed in HEK293WT, and to explore these findings further we here included binding studies with the preferred O-glycan structures on different mucin TRs (**Fig. 3a**).”

We have added data to demonstrate that the adhesins do not bind to HEK293^{WT} and HEK293^{KO}_{GCNT1/ST6GALNAC2-4} not expressing mucin TR reporters (GFP negative) in new Supplementary Figure 7b.

We have added data to demonstrate that mucin TR reporters are expressed similarly at the surface of transfected cells using anti-FLAG mAb in new Supplementary Figure 7a.

We have omitted reference to HEK293^{KO}_{GCNT1} in legend to Figure 3 – thank you!

We have added the following to the Results p.4:

“The mucin transmembrane constructs were expressed transiently in glycoengineered human embryonic kidney HEK293 cells that do not endogenously express mucins, and the secreted constructs were expressed stably^{15,24}.”

We have added the following to the Results p.7:

“We compared expression of TRs with core2 O-glycans (HEK293^{WT}) and mSTa O-glycans (HEK293^{KO}_{GCNT1/ST6GALNAC2/3/4}) (**Fig. 3a**), which replicated highly select mucin TR binding specificities, and also surprisingly uncovered that O-glycosylation of some mucin TRs do not follow the general glycosylation capacities. Importantly, membrane expression levels were comparable between mucin TR reporters and between isogenic cell lines as confirmed by anti-FLAG staining (**Supplementary Figure 7a**).”

Query #6: A number of groups have previously studied the O-glycopeptidase StcE: enzymatic activity, substrate specificity, structural studies showing an inhibitory role for sialyl-Tn in its activity, and that this reagent can be used as a lectin following mutation. The current study expands on this using the glycoengineered mucin reagents. The high/differential binding to pancreatic and breast tumors using X409 is novel and potentially important. There data seem to be part of Supplemental Data only.

Response #6: We agree that one major contribution of our studies of StcE is the discovery of the X409 module that confers the mucin-binding properties, and the essential part of this is presented in Figure 5. We

do not agree that previous studies have shown that certain glycoforms prevent StcE cleavage. In fact the original structural study of StcE concluded that larger core 1 and core 2 structures were well accommodated in docking studies and sugar residues attached at C6 of the core α -GalNAc moiety flanked the protein surface adjacent to the active site⁴⁸. We would be very interested in learning if other data exist as to the inhibitory effect of STn as indicated.

Action #6: The labelling on Figure 5 has been improved for clarity, and we now include data with the StcE catalytic mutant to further confirm that the mucin binding properties of StcE is solely ascribable to the X409 domain.

Query #7: The last figure shows that Influenza A can bind sialylated epitopes on mucins. It is not clear here if the binding depends on the simple expression of the glycan (which may be differentially expressed on various mucin TRs) or if there are other mucin/TR specific effects. Wouldn't similar binding not be observed if the biosensor was coated with synthetic glycans in the absence of the mucin? The reason for including these data in this manuscript is not entirely clear.

Response #7: This must be a misinterpretation of the experimental design and results. The study provides another illustration for use of defined mucin TRs and provides confirmation that PR8 selectively binds to sialylated core2 O-glycans on MUC1 and not other mucins with such glycans. High density of glycans can be used to demonstrate binding e.g. in glycan arrays, but clearly the selectivity for MUC1 is an important finding. We modified the text as follows:

Action #7: We have reworded the following text p.10 in Results:

“Here, we addressed if the sequence and glycosylation of four mucin TRs produced in HEK293^{WT} or HEK293^{KO GCNT1} influenced IAV HA-NA activity by analyzing binding and dissociation kinetics of the laboratory mouse-adapted influenza strain A/Puerto Rico/8/1934 (H1N1) (PR8) virions (α 2-3Neu5Ac linkage specificity) to loaded sensors using biolayer interferometry^{56,57}. Biolayer interferometry enables the interactional study of the PR8 virions with glycans presented in their genuine protein-linked form.”

and

“Next, the different mucin TR reporters were assessed for their ability to compete with PR8 virion binding to sensors coated with recombinant soluble lysosomal-associated membrane glycoprotein 1 (LAMP1), which carries multiple N-glycans with α 2-3 and α 2-6 linked sialic acids and serves as a potent IAV receptor⁵⁸.”

and

“Self-elution was slowest from MUC1-coated sensors, remarkably despite the fact that the MUC1 TR reporter contains fewer O-glycans compared to the other TR reporters. Self-elution from mucin TRs with core1 O-glycans was faster compared to their counterparts with core2 O-glycans and still slowest release were observed for MUC1 TR reporter (**Fig. 6a**).”

REVIEWERS' COMMENTS

Reviewer #1 (Remarks to the Author):

This is as before an important manuscript. The text and the figures are now much better with a great discussion. Only a few minor points:

MINOR TEXT CHANGES

1. The lab. slang 'construct' is still present at three places and should be replaced by 'plasmid' in line 104, 123, and 258.
2. Other use PTS instead of TR. Add '(also called PTS sequences)' in line 85.
3. Line 130-132: Replace with: 'Most of the mucins contain multiple TR sequences and for the mucins MUC2, MUC3, MUC5B, and MUC6 multiple TR reporters were expressed and analyzed.'
4. Remove paragraph 'Data analysis' at line 674 as the same text is already present in line 599.
5. Supplementary Table 1. The authors have used Uniprot sequences, but unfortunately the sequences for MUC2 and especially MUC6 are incomplete in Uniprot. A correct version can be found in Ref: Svensson, F., Lang, T., Johansson, M. E. V., and Hansson, G. C. The central exons of the human MUC2 and MUC6 mucins are highly repetitive and variable in sequence between individuals. *Scientific Reports* 8, 17503. 2018. Only replace the amino acid numbers and there is no need to replace anything in the main text.

Reviewer #2 (Remarks to the Author):

I have no further comments. The authors have appropriately responded to my questions and comments.

Reviewer #3 (Remarks to the Author):

The authors have responded to my critique adequately. The revised manuscript is well done. Congratulations!

Point-by-point response & action list to reviewer comments

Reviewer #1

Query #1: The lab. slang 'construct' is still present at three places and should be replaced by 'plasmid' in line 104, 123, and 258.

Response #1: Thank you.

Action #1: We have now removed or changed the word 'construct' to reporter protein.

Query #2: Other use PTS instead of TR. Add '(also called PTS sequences)' in line 85.

Response #2: Agree, and we prefer to use TR throughout the text to make it clear to the reader that the reporters contain sequences from the repeat regions.

Action #2: line 78 (previous 88) now states:

"A common characteristic of all mucins is that the major part of their extracellular region is comprised of variable number of imperfect tandem repeated (TR) sequences (also called PTS sequences) that carry dense O-glycans (Fig. 1), with the notable exception of MUC16 that contains a large, densely O-glycosylated N-terminal region without TRs".

Query #3: Line 130-132: Replace with: 'Most of the mucins contain multiple TR sequences and for the mucins MUC2, MUC3, MUC5B, and MUC6 multiple TR reporters were expressed and analyzed.'

Response #3: Thank you

Action #3: Line 124-125 have been updated with the sentence above.

Query #4: Remove paragraph 'Data analysis' at line 674 as the same text is already present in line 599.

Response #4: Thank you for spotting this.

Action #4: Corrected.

Query #5: Supplementary Table 1. The authors have used Uniprot sequences, but unfortunately the sequences for MUC2 and especially MUC6 are incomplete in Uniprot. A correct version can be found in Ref: Svensson, F., Lang, T., Johansson, M. E. V., and Hansson, G. C. The central exons of the human

MUC2 and MUC6 mucins are highly repetitive and variable in sequence between individuals. Scientific Reports 8, 17503. 2018. Only replace the amino acid numbers and there is no need to replace anything in the main text.

Response #5: Thank you. Uniprot has not updated their sequences for MUC2 and MUC6. Therefore we added this sentence in Supplementary Table 1.

Action #5: Supplementary Table 1 now states:

“The amino acid number is derived from the incomplete version of human MUC2 and MUC6 sequence Uniprot Q02817 and Q6W4X9. A complete sequence of MUC2 and MUC6 can be found in Frida Svensson et al. (2018) Sci Reports 8, 17503.”